# A global assessment of marine heatwaves and their drivers

Neil J. Holbrook [1,2], Hillary A. Scannell [3], Alexander Sen Gupta [4,5], Jessica A. Benthuysen [6], Ming Feng [7], Eric C.J. Oliver [1,2,8], Lisa V. Alexander[4,5], Michael T. Burrows [9], Markus G. Donat [5,10], Alistair J. Hobday [11], Pippa J. Moore [12], Sarah E. Perkins-Kirkpatrick[4,5], Dan A. Smale [13,14], Sandra C. Straub [14] & Thomas Wernberg [14]

Marine heatwaves (MHWs) can cause devastating impacts to marine life. Despite the serious consequences of MHWs, our understanding of their drivers is largely based on isolated case studies rather than any systematic unifying assessment. Here we provide the first global assessment under a consistent framework by combining a confidence assessment of the historical refereed literature from 1950 to February 2016, together with the analysis of MHWs determined from daily satellite sea surface temperatures from 1982–2016, to identify the important local processes, large-scale climate modes and teleconnections that are associated with MHWs regionally. Clear patterns emerge, including coherent relationships between enhanced or suppressed MHW occurrences with the dominant climate modes across most regions of the globe – an important exception being western boundary current regions where reports of MHW events are few and ocean-climate relationships are complex. These results provide a global baseline for future MHW process and prediction studies.

[1] Institute for Marine and Antarctic Studies, University of Tasmania, Hobart 7001 Tasmania, Australia. [2] Australian Research Council Centre of Excellence for Climate Extremes, University of Tasmania, Hobart 7001 Tasmania, Australia. [3] School of Oceanography, University of Washington, Seattle 98105 WA, USA. [4] Climate Change Research Centre, The University of New South Wales, Sydney 2052, Australia. [5] Australian Research Council Centre of Excellence for Climate Extremes, The University of New South Wales, Sydney 2052 New South Wales, Australia. [6] Australian Institute of Marine Science, Townsville 4810 Queensland, Australia. [7] CSIRO Oceans and Atmosphere, Indian Ocean Marine Research Centre, Crawley 6009 Western Australia, Australia. [8] Department of Oceanography, Dalhousie University, Halifax, NS B3H 4R2, Canada. [9] Scottish Association for Marine Science, Scottish Marine Institute, Oban, Argyll, Scotland PA37 1QA, UK. [10] Barcelona Supercomputing Center, Barcelona 08034, Spain. [11] CSIRO Oceans and Atmosphere, Hobart 7000 Tasmania, Australia. [12] Institute of Biological, Environmental and Rural Sciences, Aberystwyth University, Aberystwyth SY23 3DA, UK. [13] Marine Biological Association of the United Kingdom, The Laboratory, Citadel Hill, Plymouth PL1 2PB, UK. [14] UWA Oceans Institute and School of Biological Sciences, The University of Western Australia, Crawley 6009 Western Australia, Australia. Correspondence and requests for materials should be addressed to N.J.H. (email: neil.holbrook@utas.edu.au)

Extended periods of anomalously warm ocean temperatures can have major impacts on marine biodiversity and eco-systems[1–3] and the economics of regional fisheries[4,5]. Despite a growing appreciation of their importance, scientific understanding of marine heatwaves (MHWs) is in its infancy compared to that of atmospheric heatwaves. In particular, there is limited understanding of the physical processes that give rise to MHWs[6,7], and how large-scale climate variability modulates the likelihood and severity of these events.

In recent years, a number of prominent MHWs have been reported with devastating changes to marine ecosystems around the globe. These include MHW events in the northwest Atlantic in 2012[4], northeast Pacific Ocean from 2013–2016[8], Tasman Sea in 2015/16[9], and waters around tropical Australia in 2015/16[10]. On a global scale, these extreme temperature events have increased in frequency[11], a trend projected to continue throughout the twenty-first century[12], and hence there is a pressing need to understand their physical drivers.

The focus of this study was to develop and apply a consistent framework for understanding how regional MHW events relate to different modes of climate variability and the physical conditions causing waters to warm above a threshold. This framework aimed to inform the predictability of MHW events and, on a global scale, our capability to detect and understand how MHWs emerge. A useful lens for understanding the formation, main-tenance, and decay of MHWs is the upper ocean mixed layer heat budget, which includes the local processes responsible for changes in surface ocean temperatures. For MHW events, the dominant contributions to the temperature tendency within the mixed layer[13–15] are given by

$$\frac{\partial T}{\partial t} = -\frac{1}{H}\int_{-H}^{0} (\mathbf{u} \cdot \nabla_h T)dz + \frac{Q}{\rho C_p H} + \text{residual} \qquad (1)$$

where the first term on the right hand side (RHS) is the hor-izontal advection (ocean advective fluxes) owing to the vertical integral of the product between the depth-dependent horizontal velocity vector $\mathbf{u}$ and the horizontal gradient of the mixed layer temperature $T$, the second term is the contribution from the net air-sea heat flux $Q$, where $\rho$ is the average seawater density, $C_p$ is the specific heat capacity of seawater (4000 J kg$^{-1}$ °C$^{-1}$), $H$ is the mixed layer depth, and the residual third term includes horizontal eddy heat fluxes and the heat flux at the bottom of the mixed layer owing to radiative heat loss, vertical diffusion, entrainment, and vertical advection. To generate MHWs, the physical pro-cesses on the RHS of Equation (1) have to contribute a sufficiently large net positive temperature tendency to increase the surface ocean temperature over a threshold[6]. Oceanic advective fluxes include unusually intense poleward flow associated with the transport of warmer waters, enhanced heat advection by boundary currents, or reduced turbulent mixing. Favourable air-sea heat fluxes for MHWs include contributions from anomalous net downward surface radiation characteristics of reduced cloud cover, and from convective air-sea heat flux and suppressed latent heat loss from anomalously weak surface winds. While Equation (1) explicitly describes temperature variations that extend throughout the mixed layer, MHWs may extend deeper. The expression of surface intensified MHWs may also depend on the state of the subsurface ocean, such as shoaling of the mixed layer base[10,15].

In the majority of research conducted to date, and in our global analysis here, MHWs have been identified and characterized based on sea surface temperature (SST). This focus reflects the relative scarcity of subsurface ocean data, and the reported eco-logical impacts that are prevalent in the upper ocean where biological productivity is greatest. Key mechanisms that drive temperature changes in the mixed layer include local internal variability (e.g. eddy instabilities) or large-scale modes of climate variability that act to modulate the local conditions—either from local sources, for example, extreme ocean-atmosphere coupled feedbacks in the eastern equatorial Pacific during extreme El Niño-Southern Oscillation (ENSO) events, or remote sources via teleconnection mechanisms, for example, the propagation of planetary waves in the atmosphere (that give rise to distant changes in the surface winds, cloud cover, etc.) or ocean (that can change the depth of the thermocline and drive remote cir-culation changes).

We established an analytical framework to examine the influence of these drivers on MHWs regionally. Our first goal was to identify regions where robust statistically significant relationships exist between surface signatures of MHW occur-rence and large-scale climate modes. We recognize and caution that significant statistical relationships do not necessarily indi-cate causal links. However, these analyses are indicative of potentially important connections that can provide clear pointers to potential predictability. Importantly, the relationships we have found and will highlight in the results between large-scale cli-mate modes and MHW occurrences do not simply mirror those between climate modes and SST anomalies (SSTA) more generally.

Our second goal was to perform a comprehensive and sys-tematic global synthesis of the existing peer-reviewed literature on historical MHWs (from 1950 to February 2016) that have occurred in oceanographically and/or climatically distinct regions. We conducted a confidence assessment of their char-acteristic drivers and processes to determine: the processes in the mixed layer temperature budget (RHS of Equation 1) important for MHW formation; climatological systems and their config-urations (e.g. high/low pressure systems, jet stream location), and teleconnection processes (e.g. atmospheric or oceanic Kelvin and Rossby waves), that either force or modulate the temperature budget terms; and large-scale modes of climate variability that have played a role in modulating individual identified MHWs, either locally or via teleconnections.

Our third goal was to revisit and analyze these reported events since 1982 by producing consistent measurements of the key MHW characteristics based on daily satellite SST data. These characteristics were estimated based on a consistent MHW defi-nition[6], which allows for comparability since the MHWs identi-fied in our literature search have been reported using a diverse set of criteria.

Our study concludes that the consideration of extremes in an ocean mixed layer temperature tendency budget framework provides a useful lens for understanding the formation, main-tenance, and decay of surface MHWs locally. Furthermore, remote influences (e.g. climate modes), and ocean/atmosphere teleconnection processes, appear to be important modulators for the enhancement or suppression of MHW occurrences and thus their likelihoods. We expect these findings to be beneficial to studies on the potential predictability of MHWs and provide future direction for prediction studies.

## Results

**Outline**. The key results from our assessment of MHW drivers are provided under four broad ocean-climate zones—tropical latitudes, middle and high latitudes, western boundary currents and extension regions, and eastern boundary currents—with the synthesis review provided in Supplementary Notes 1–5, and MHW event summaries in Supplementary Tables 4–20. Within these four zones, a total of 30 MHWs spanning 22 case-study regions (Fig. 1a) were identified based on our search criteria

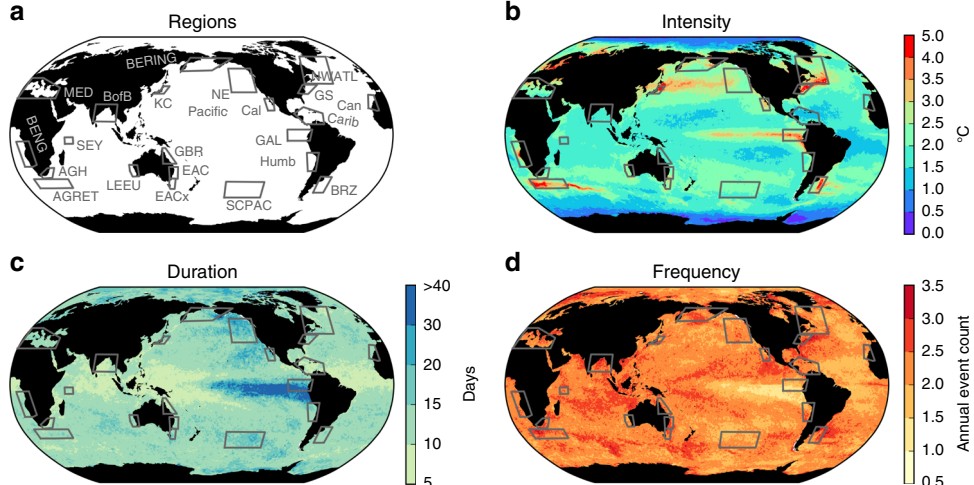

**Fig. 1** Global MHW characteristics and case-study regions. 34-year (1982–2015) average properties of MHWs based on application of the MHW definition[6] to daily sea surface temperatures from the NOAA OI SST V2 dataset across the globe. **a** A total of 22 case-study regions investigated. The spatial distribution of MHW properties (0.25° × 0.25° resolution) here includes **b** annual mean intensity (°C), **c** duration (days), and **d** frequency (event counts per year). The four ocean-climate zones are the tropical latitudes, middle and high latitudes, western boundary currents and their extensions (WBCs), and eastern boundary currents (EBCs). These case-study regions are listed as follows: tropical latitudes [Great Barrier Reef (GBR); Seychelles Islands (SEY); Galapagos Islands (GAL); Bay of Bengal (BofB); Caribbean Sea (Carib)], middle and high latitudes [Mediterranean Sea (MED); Bering Sea (BERING); northwest Atlantic (NWATL); northeast Pacific (NE Pacific); south central Pacific (SCPAC)], WBCs [Gulf Stream (GS); Kuroshio Current (KC); Brazil-Malvinas Confluence (BRZ); Agulhas Current (AGH); Agulhas Retroflection (AGRET); East Australian Current (EAC); East Australian Current Extension (EACx)], and EBCs [Benguela Current (BENG); Leeuwin Current (LEEU); Baja California (Cal); Iberian/Canary Current (Can); Humboldt/Peru Current (Humb)]

(Supplementary Table 1). These criteria were consistent with the qualitative definition of a MHW as a 'discrete prolonged anomalously warm water event' at a particular location[6].

Our findings are categorized and summarized by case-study region and time scale in Table 1, and a confidence level around the climate modes or local processes important in modulating specific events has been assigned based on expert assessment of the number of studies and agreement across those studies identified from the literature (Supplementary Fig. 2). Similar to atmospheric heatwaves, we found that MHWs may be caused by the interaction of many local and remote processes and phenomena acting across a large range of temporal and spatial scales (Fig. 2). A global composite analysis (Fig. 3) summarizes the statistical relationships between the important large-scale climate modes and increased MHW occurrences (or likelihood) within the case-study regions and elsewhere. This statistical analysis of data provides an additional unifying perspective that complements our literature synthesis and confidence assessment.

It is evident that the magnitude, frequency, and duration of MHW events have a heterogeneous distribution across the global oceans (see Fig. 1). Except for the upwelling regions of the equatorial Pacific cold tongue, the highest intensity MHWs (i.e. with largest SSTA) are primarily associated with boundary currents like the Gulf Stream or their extensions. Based on our literature search and synthesis (Supplementary Notes 1–5 and Supplementary Tables 3–20), we found it somewhat surprising that there have been very few reported studies of MHWs in the western boundary current regions, despite the large temperature variability found there. In contrast, a greater richness of information has been reported on MHWs in eastern boundary current regions. On average, the longest-duration MHWs tend to occur in the eastern equatorial Pacific associated with El Niño events, aside from a few longer individual events noted elsewhere in recent years, e.g. the recent 2013–2016 northeast Pacific Blob event[8] and 2015/16 Tasman Sea MHW[9].

**Statistical analysis of climate modes and MHW occurrences**. To better frame and contextualize our global assessment of the regional literature on MHW processes and drivers, we first characterized the important relationships between large-scale climate modes and MHW occurrences. This was necessary since individual case studies described in the literature used different methods to define and characterize MHW events. This non-uniformity makes it problematic to directly compare multiple events and their drivers in a consistent manner. To address this, we used satellite SST data following the established MHW definition[6] and adapted for strong events to uniformly characterize event duration, maximum intensity, and maximum spatial extent of the individual MHWs reported in the literature from 1982–2016 (Table 2). The application of a consistent MHW definition allowed us to compare events and statistically analyze MHW occurrences in relation to large-scale climate modes of variability. Our approach provided a coherent and useful framework to evaluate and confirm whether the warm ocean temperature events identified in our literature search were indeed MHWs, thus providing a more robust confidence assessment.

Different phases of known climate modes are associated with enhanced or suppressed likelihoods of MHWs[16]. We quantified this effect through a composite analysis of MHW occurrences during positive and negative phases of each climate mode identified as being a driver of MHWs, based on analysis of satellite SST data since 1982 (see Methods). We summarize which climate modes (and phases) have the largest statistically significant association with an increase (Fig. 3) or decrease (Supplementary Fig. 5) in occurrence of MHWs at each grid cell across the global ocean. Often, there is more than one (sometimes multiple) climate mode that has a significant relationship with increased MHW occurrences at a given location and/or case-study region (Fig. 4).

The relationships between MHW occurrences and climate modes (Fig. 3) are complex, although the large-scale patterns are broadly

**Table 1 Characteristic MHW drivers from literature assessment**

| Typology | Case study | Timescale | | | | | | | |
| --- | --- | --- | --- | --- | --- | --- | --- | --- | --- |
| | | Synoptic | | Seasonal to intraseasonal | | Interannual | | Decadal | |
| | | Mode/teleconnection | Local process | Mode/teleconnection | Local process | Mode/teleconnection | Local process | Mode/teleconnection | Local process |
| EBC | Benguela | | | ABF, RWS, KWO, $MJO_1$ | ADV, $ASHF_1$ | RWS, $KWO_1$ | ADV, $VP_1$ | | |
| | Leeuwin | | | RASC, SLP(−), $LWS_1$ | ADV, EHF, ASHF, $VP_1$ | ENSO(−), SLP, $RWS_1$ | ADV, $ASHF_1$ | PDO(−), $ENSO_1$ | $ASHF_1$ |
| | Baja California | | | $LWS_2$ | ADV, ASHF, $VP_1$ | ENSO(+), RWS, SLP$(-)_2$ | ASHF, VP, $ADV_1$ | | |
| | Iberian/Canary | $AB_4$ | $ASHF_4$ | NAO(−), RASC, $RWS_4$ | ADV, $ASHF_4$ | $JS_4$ | $ASHF_4$ | | |
| | Humboldt/Peru | | | KWO, $RWS_2$ | VP, ADV, $ASHF_2$ | ENSO(+), $RWS_2$ | ADV, $ASHF_2$ | | |
| WBC | *Gulf Stream | $JS_5$ | ASHF, ADV, $EHF_5$ | JS, NAO$(+)_5$ | ASHF, ADV, EHF, $VP_5$ | | | $AMO_5$ | ADV, $EHF_5$ |
| | * Kuroshio | $RASC_5$ | $ASHF_5$ | | | ENSO, $RWO_5$ | ADV (fronts), $EHF_5$ | PDO, PNA, AL, RWA, $RWS_5$ | $ASHF_5$ |
| | * Brazil-Malvinas Confluence | RASC, RWS, $SLP_5$ | ASHF, $ADV_5$ | RASC, $SLP_5$ | ASHF, ADV, $EHF_5$ | | | | |
| | * Agulhas | | | RASC, SLP, $RWS_5$ | ADV, VP, $EHF_5$ | ENSO(−), IOD$(-)_5$ | ASHF, ADV, $VP_5$ | PDO$(-)_5$ | $ASHF_5$ |
| | Agulhas Retroflection | | | RWS, $SLP_5$ | $ADV_5$ | | | | |
| | * East Australian Current | | | $RASC_3$ | ADV, VP, $EHF_3$ | ENSO(+), $CPEN_3$ | ADV, EHF, $VP_3$ | PDO/$IPO_3$ | ASHF, ADV, $EHF_3$ |
| | East Australian Current Extension | | | $BI_5$ | ADV, $EHF_5$ | $ENSO_3$ | ADV, $ASHF_3$ | | |
| Tropics | Great Barrier Reef | AB, RASC, $LWS_3$ | ASHF, $ADV_3$ | | | ENSO(+), $CPEN_1$ | VP, ADV, $ASHF_1$ | PDO/IPO, $RWS_1$ | ASHF, $EHF_1$ |
| | Seychelles Is. | | | RASC, SLP, $RWS_3$ | ADV, $VP_1$ | IOD$(+)_2$ | ASHF, ADV, $EHF_1$ | | |
| | Galápagos Is. | | | $RWS_1$ | $VP_1$ | ENSO( + ), $RWS_1$ | VP, ADV, $ASHF_1$ | PDO$(+)_4$ | ASHF, $TD_2$ |
| | Bay of Bengal | RASC, $SLP_4$ | ASHF, $TM_4$ | RASC, $ASM_4$ | ASHF, $ADV_4$ | $IOD_4$ | $ASHF_4$ | | |
| | Caribbean Sea | | | | | ENSO$(+)_5$ | $ASHF_5$ | | |
| MHL | Mediterranean Sea | SLP(+), LWS, $RWA_1$ | ASHF, $VP_1$ | | | | | $AMO_1$ | $ASHF_1$ |
| | Bering Sea | SLP$(+)_1$ | $ASHF_1$ | AL, $LWS_1$ | ASHF, $ADV_1$ | ENSO(+), RWA, SLP$(+)_1$ | $ASHF_1$ | AL, $PDO_1$ | $ASHF_1$ |
| | Northwest Atlantic | | | JS, RASC, RWS, $SLP_1$ | $ASHF_1$ | | | | |
| | Northeast Pacific | SLP(+), $LWS_1$ | ASHF, $EHF_1$ | $AL_1$ | $ASHF_1$ | ENSO(+), $RWA_1$ | AL, SLP$(+)_1$ | NPO(+), RWS, NPGO$(+)_1$ | SLP$(+)_1$ |
| | South Central Pacific | | | | | ENSO(+), $RWA_5$ | ASHF, $ADV_5$ | | |

| Large-scale and regional climate modes | | Teleconnection processes and climatological features | | Local processes affecting the mixed layer temperature budget | |
| --- | --- | --- | --- | --- | --- |
| ENSO(+/−) | El Niño-Southern Oscillation | AB | Atmospheric Blocking | ADV | Ocean Advection |
| CPEN | Central Pacific El Niño | AL | Aleutian Low | EHF | Eddy heat flux |
| IPO | Interdecadal Pacific Oscillation | SLP(+/−) | Sea Level Pressure | ASHF | Air-sea heat flux |
| PDO(+/−) | Pacific Decadal Oscillation | JS | Jet Stream position | VP | Vertical Processes (entrainment, turbulent mixing, thermocline deepening) |
| IOD(+/−) | Indian Ocean Dipole | PNA | Pacific North American Pattern | | |
| MJO | Madden-Julian Oscillation | RWA | Rossby Wave (Atmospheric) | | |
| NAM | Northern Annular Mode | ABF | Angola-Benguela Front | | |
| NAO(+/−) | North Atlantic Oscillation | BI | Baroclinic Instability | | |
| NPGO(+/−) | North Pacific Gyre Oscillation | KWO | Kelvin Wave (Oceanic) | | |
| NPO | North Pacific Oscillation | RWO | Rossby Wave (Oceanic) | | |
| AMO | Atlantic Multi-decadal Oscillation | RWS | Regional wind stress change | | |
| SAM | Southern Annular Mode | RASC | Regional air-sea coupling | | |
| ASM | Asian Summer Monsoon | LWS | Local wind stress change | | |

Our literature assessment of MHW drivers characterizes the contributions from: large-scale or regional climate modes (e.g. ENSO), atmospheric or oceanic teleconnection processes and climatological features (e.g. Rossby waves, fronts) [Mode/Teleconnection columns]; and local processes (e.g. ocean advection) affecting the MHW heat budget [Local Process columns] across four time scales (synoptic, intraseasonal, interannual and decadal) and classified by typology, i.e. eastern boundary currents (EBC), western boundary currents and extensions (WBC), tropics, and middle and high latitudes (MHL). For regions where no drivers or processes could be identified from the literature, the box is left blank. For ENSO, PDO, IOD, NAO and SLP, individual studies may indicate whether MHWs are associated with positive (+) or negative (−) phase. Case studies with an asterisk (*) have no documented MHWs, but literature identifies processes and modes that cause changes in the mixed layer temperature budget. Numbers correspond to a qualitative confidence assessment for literature documented MHW mode/teleconnection and local processes respectively. Confidence ratings are explained in the Methods and Supplementary Fig. 2, and include very high ($_1$), high ($_2$), medium ($_3$), low ($_4$), and very low ($_5$) confidence ratings. Corresponding references are provided in Supplementary Table 1

consistent with known SSTA patterns (note that there are clear differences, e.g. compare the regressed SSTA against the composite MHW occurrence plots in Supplementary Figs. 11 and 14 for the North Atlantic Oscillation (NAO) and Southern Annular Mode (SAM)). Some characteristics include the relationship between ENSO and the extensive area of influence in the Pacific basin, with El Niño apparently enhancing (and La Niña suppressing) MHW occurrences in the central and eastern Pacific Ocean and suppressing MHW occurrences in a characteristic chevron

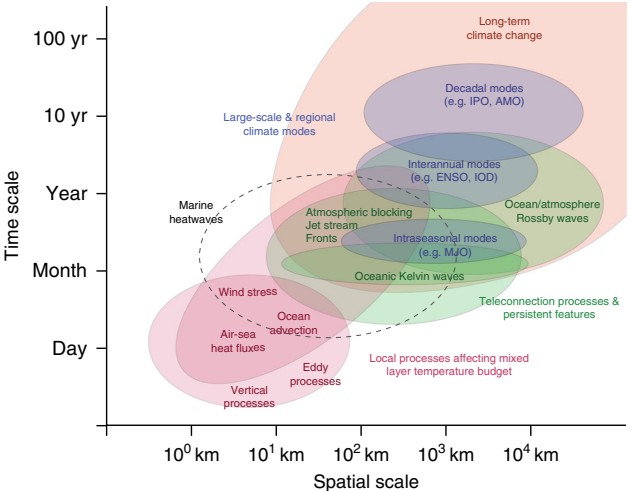

**Fig. 2** Space and time scales of characteristic MHW drivers. Schematic identifying the characteristic marine heatwave drivers and their relevant space and time scales. Included are drivers that force locally (through processes affecting the mixed layer temperature budget (red)), and those that act to modulate MHW occurrences from regional or remote sources (climate modes (blue)) via atmospheric and/or oceanic teleconnection processes (green). Each driver is mapped to their relevant time and spatial scales identified from a synthesis of information contained in the literature. The black dashed line outlines the typical scales for MHWs

shaped-region of the western Pacific Ocean extending eastwards into higher latitudes (again with opposite signal for La Niña). However, Fig. 3 indicates that MHW occurrences in the central Indian Ocean are also significantly related to ENSO (likely via the ENSO-driven Indian Ocean Basin-wide mode (IOB)[17]), as well as in parts of the Southern Ocean and the eastern Atlantic basin. This highlights the prominence of ENSO influence for many regions around the world. Similarly, the central Pacific (CP) El Niño is strongly related to MHWs in the central and western Pacific. We also note a stronger association between MHW occurrences off the west coast of Australia—where Ningaloo Niños occur[18] and CP El Niño (represented in Fig. 3 by El Niño Modoki (EMI)), rather than with the canonical ENSO[14]. At higher latitudes in the Pacific, where we might expect longer time scale variations associated with ocean memory to play a role[19], the lower frequency Interdecadal Pacific Oscillation (IPO) has the strongest association with MHWs. The North Pacific Gyre Oscillation (NPGO) and Pacific Decadal Oscillation (PDO) have been found to influence temperature and sea level changes in the North Pacific Subtropical Gyre[20] and specifically the 2013–16 northeast Pacific MHW[8].

The Indian Ocean Dipole (IOD) dominates in the western and eastern tropical Indian Ocean, also apparently influencing much of the Maritime Continent (Fig. 3b). The North Atlantic Ocean is dominated by the NAO, which has two centres of activity in the far north and in the tropical North Atlantic Ocean. Between these two regions there is no clear dominant driver. In the eastern South Atlantic Ocean, MHWs are apparently most strongly modulated by the Atlantic Niño. In the western basin, Pacific drivers including the IPO (identified by the Tripole index, see Methods), NPGO and CP El Niño, appear to be important. No clear climate mode dominates over the Southern Ocean, although there are some extended regions affected by SAM, ENSO, and the IPO.

While Fig. 3b highlights which climate mode has the strongest relationship with MHW occurrence at a given location, in many instances there are statistically significant relationships with multiple climate modes. This can be seen in Fig. 4 shown as all the statistically significant relationships between MHW

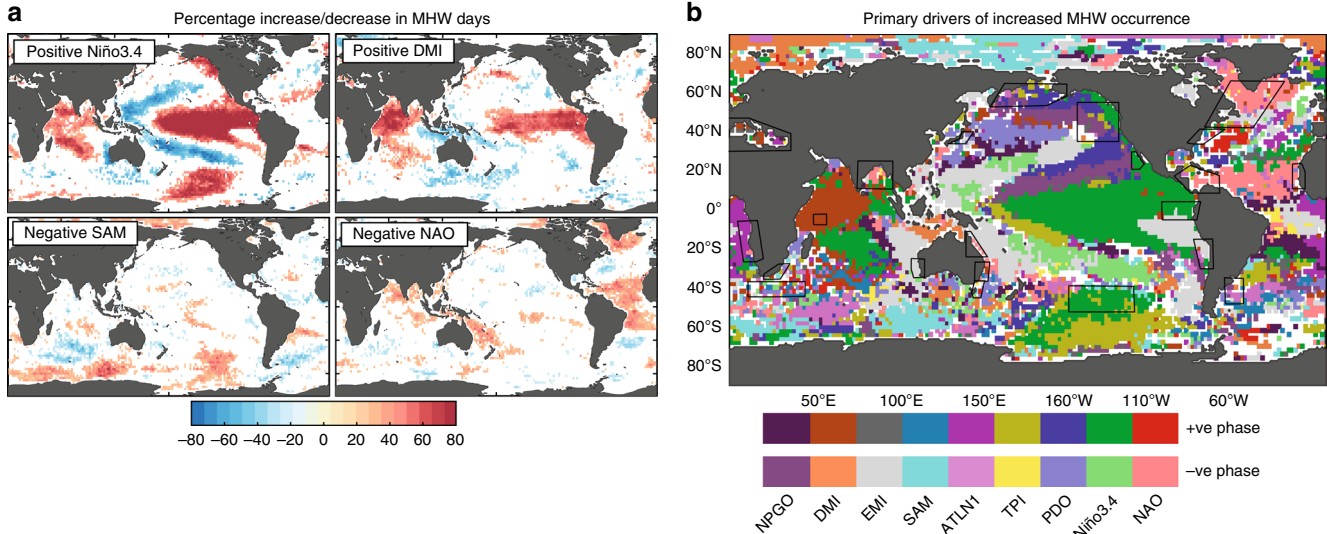

**Fig. 3** Links between enhanced or suppressed MHW occurrences and climate modes. **a** The percentage of days in which MHWs increase or decrease during a phase of four climate modes. **b** Summary schematic showing the locations where climate modes (and phases) have the greatest significant impact on enhancing or suppressing the number of MHW days (see Methods). The percentage of enhancement or suppression for individual climate modes used to construct this schematic are shown in Supplementary Figs. 7–15. When there are no statistically significant relationships between known climate modes and MHW occurrences, the areas are shaded in white

**Table 2 Characteristics of strong MHWs identified in the literature (1982–2016)**

| Typology | Case study | Marine Heatwaves (Refs. in SI) | Metrics | | | |
| --- | --- | --- | --- | --- | --- | --- |
| | | | Start date (>98%) | End date (<98%) | Max intensity [°C (Date)] | Max area >98% [Mkm² (Date)] |
| EBC | Benguela | 1982/83 | 21/12/1982 | 31/1/1983 | 4.8 (22/1/1983) | 1.4 (24/1/1983) |
| | | 1984 | 8/3/1984 | 12/3/1984 | 4.6(9/3/1984) | 0.13(9/3/1984) |
| | | 1995 | 5/2/1995 | 19/2/1995 | 4.7(7/2/1995) | 0.2 (17/2/1995) |
| | | | 21/3/1995 | 1/4/1995 | 5.7(22/3/1995) | 0.3 (27/3/1995) |
| | | | 11/4/1995 | 21/4/1995 | 6.7(16/4/1995) | 0.6 (17/4/1995) |
| | | 2001 | 27/4/2001 | 20/5/2001 | 5.3 (14/5/2001) | 0.7 (16/5/2001) |
| | Leeuwin | 1989 | – | – | – | – |
| | | 1999/2000 | – | – | – | – |
| | | 2011 | 8/2/2011 | 23/3/2011 | 6.8(26/2/2011) | 0.95(4/3/2011) |
| | | 2012 | 20/1/12 | 29/1/12 | 5.1(28/1/11) | 0.3 (17/1/11) |
| | | 2012/13 | 2/1/2013 | 10/1/2013 | 4.0 (2/1/2013) | 0.24 (5/1/2013) |
| | Baja California | 1982/83 | 10/2/1983 | 5/3/1984 | 4.0(23/2/1983) | 0.12(24/2/1983) |
| | | 1997 | 14/11/1997 | 11/12/1997 | 4.6(26/11/1997) | 0.37(6/12/1997) |
| | | | 30/12/1997 | 14/1/1998 | 4.2(31/12/1997) | 0.39 (3/1/1998) |
| | | | 10/2/1998 | 20/2/1998 | 3.5(19/2/1998) | 0.17(16/2/1998) |
| | | 2015/16 | 1/7/2015 | 2/1/2016 | 6.4(12/9/2015) | 21.2(2/11/2015) |
| | Iberian/Canary | – | – | – | – | – |
| | Humboldt/Peru | 1982 | 23/8/1982 | 7/10/1982 | 4.2(10/9/1982) | 2.1(29/9/1982) |
| | | | 14/10/1982 | 18/11/1982 | 3.9(16/11/1982) | 0.25(16/11/1982) |
| | | | 26/11/1982 | 14/12/1982 | 5.5(11/12/1982) | 1.0(9/12/1982) |
| | | 1983 | 5/1/1983 | 24/7/1983 | 9.6 (6/5/1983) | 4.4 (24/6/1983) |
| | | 1997/98 | 30/5/1997 | 18/6/1997 | 4.1(2/6/1997) | 5.4(12/6/1997) |
| | | | 9/7/1997 | 25/8/1997 | 6.2(21/7/1997) | 4.7(18/8/1997) |
| | | | 3/9/1997 | 17/9/1997 | 6.5(7/9/1997) | 8.1 (4/9/1997) |
| | | | 24/9/1997 | 17/11/1997 | 6.2(7/11/1997) | 7.0(26/9/1997) |
| | | | 25/11/1997 | 28/12/1997 | 6.9(28/11/1997) | 4.1(5/12/1997) |
| WBC | East Australian Current Extension | 2016 | 7/12/2015 | 12/4/2016 | 7.5(8.2.2016) | 0.7(6/3/2016) |
| | Agulhas Retroflection | 1985 | 29/11/1985 | 5/12/1985 | 0.2(2/12/1985) | 5.7(29/11/1985) |
| Tropics | Great Barrier Reef/Coral Sea | 1998 | 9/2/1998 | 18/3/1998 | 3.4(4/3/1998) | 0.14 (5/3/1998) |
| | | 2001/02 | 30/12/2001 | 15/1/2002 | 3.3 (6/1/2002) | 0.5(8/1/2002) |
| | | 2015/16 | 28/2/2016 | 4/4/2016 | 4.0 (15/3/2016) | 2.6 (12/3/2016) |
| | Seychelles Is. | 1997/98 | 14/1/1998 | 3/2/1998 | 3.7(25/1/1998) | 0.4(18/1/1998) |
| | | | 14/2/1998 | 22/2/1998 | 3.4(21/2/1998) | 0.9(18/2/1998) |
| | Galápagos Is. (Eq. Pac) | 1982/83 | 5/5/1983 | 31/7/1983 | 9.56(6/5/1983) | 4.7(5/6/1983) |
| | | 1987 | – | | | |
| | | 1992 | – | | | |
| | | 1997/98 | 14/7/1997 | 3/6/1998 | 6.9(28/11/1997) | 11(9/11/1997) |
| | Bay of Bengal | 2010 | 4/5/2010 | 6/6/2010 | 3.4(15/5/2010) | 1.1(14/5/2010) |
| | Caribbean Sea | 1982/83 | – | – | – | – |
| | | 1987/88 | – | – | – | – |
| | | 1989/90 | – | – | – | – |
| MHL | Mediterranean Sea | 1999 | – | – | – | – |
| | | 2003 | 6/9/2003 | 16/7/2003 | 5.5(14/6/2003) | 0.5(16/6/2003) |
| | | | 8/8/2003 | 4/9/2003 | 4.6 (29/8/2003) | 1.2(23/8/2003) |
| | Bering Sea | 1997 | 31/5/1997 | 27/6/1997 | 5.1(12/6/1997) | 0.05(4/6/1997) |
| | Northwest Atlantic Ocean | 2012 | 18/3/2012 | 25/3/2012 | 10.3 (23/3/2012) | 0.3(22/3/2012) |
| | | | 17/4/2012 | 28/4/2012 | 5.4(17/4/2012) | 0.3(25/4/2012) |
| | | | 19/5/2012 | 31/5/2012 | 7.5(23/5/2012) | 0.2(26/5/2012) |
| | | | 21/6/2012 | 10/9/2012 | 9.2(6.7/2012) | 0.1(1/8/2012) |
| | | | 19/9/2012 | 13/10/2012 | 6.1 (8/10/2012) | 0.3(8/10/2012) |
| | | | 31/10/2012 | 27/11/2012 | 8.1(27/11/2012) | 0.2(3/11/2012) |
| | Northeast Pacific Ocean | 2013–2015 | 14/11/2013 | 30/6/2014 | 4.8(3/6/2014) | 4.5(17/1/2014) |
| | | | 28/7/2014 | 31/8/2015 | 6.7(28/6/2015) | 11.7(15/2/2015) |
| | South Central Pacific Ocean | 2009/10 | 9/11/2009 | 2/3/2010 | 6.0(24//12/2009) | 3.9(25/12/2009) |

MHWs were identified using the quantitative definition[6] applied here to derive metrics (but based on a 98th-percentile threshold). Those MHWs identified in the literature that have a dash '–' did not meet the 98th percentile criteria although identified in the literature with the corresponding references (see Supplementary Table 2). Further, some of the closely separated MHW events defined here (e.g. Humboldt/Peru Current region in 1982, 1983 and 1997/98; and the northeast Pacific region in 2013/14 and 2014/15) would be considered a continuous MHW if a weaker threshold was used

suppression or enhancement with the phase of known climate modes averaged across the various case-study (boxed) regions. Our statistical analysis shows, for example, that the likelihood of a MHW in the northeast Pacific (e.g. the Blob) is elevated to about 17% during positive phases of the PDO and suppressed to about 5% during negative phases of the PDO. However, ENSO (measured by Niño3.4), the IPO (measured by the Tripole index), CP ENSO (measured by the EMI), and NPGO also have significant effects on MHW occurrences in the northeast Pacific. Further, Fig. 3a shows that in the equatorial and far North

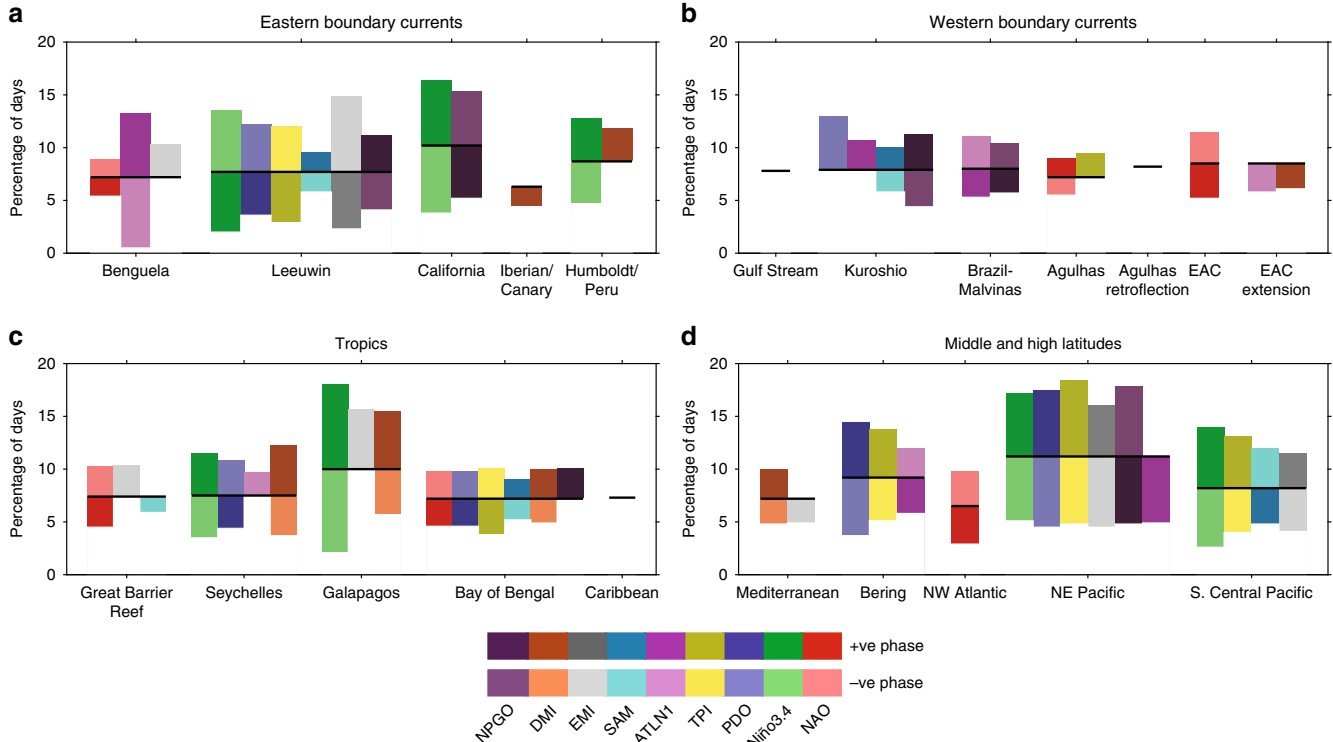

**Fig. 4** Percentage change in MHW occurrences linked to climate mode phase. The percentage of days experiencing MHWs during positive or negative phases of a climate mode, for each case-study region. The black horizontal line indicates the median percentage over the full period irrespective of the phase of the modes. Values above (below) this bar indicate the climate mode phase enhances (suppresses) the likelihood of MHW occurrences. Values are only shown when significant at the 5% level (based on a Monte Carlo sampling, see Methods). The median deviates from 10% because the MHW definition includes a 5-day threshold and the climatological period used for calculation of MHWs (1982–2012) differs from the full length of the analyzed time series (1982–2016). Climate modes associated with the various indices are described in the Methods ("Climate indices as metrics for climate mode drivers")

Atlantic Ocean, as well as the tropical southwest Pacific (Coral Sea), there are likely to be >40% more MHW days during the negative NAO phase (Supplementary Figs. 3–15 provide detail across all of the climate modes considered).

In many cases, the modulation of MHW occurrence is consistent with local changes in SSTA associated with the various climate modes. For example, the Brazil Current region is linked to local Atlantic Niño variability, the eastern North Pacific region is affected by ENSO, PDO, IPO, and NPGO, and the northwest Atlantic is primarily affected by the NAO. However, it is important to again note that significant statistical relationships do not necessarily indicate causal links. For example, Fig. 3 shows an apparent connection between the IOD and MHW suppression in the central Pacific. This link is most likely a consequence of the correlation between the Dipole Mode index (DMI) and Niño3.4 indices, which is due to the IOD being at least partly driven by ENSO variability. Interestingly, in many of the western boundary current (WBC) regions, such as the Gulf Stream, East Australian Current, and the Agulhas Current, we find no clear relationship between MHW occurrence and any of the climate modes. This implies that the key drivers of MHWs in these regions may be either persistent changes in local or remote forcing (e.g. wind stress) not associated with a large-scale climate mode(s), lagged responses to climate modes, or internal variability.

**Assessment of the literature on drivers and processes.** Based on our global confidence assessment of the historical peer-reviewed literature on the local to large-scale mechanisms that cause MHWs, we have collectively determined that surface MHWs are

the direct result of local-scale processes acting within the mixed layer (e.g. ocean heat advection, air-sea interaction or vertical mixing; Supplementary Fig. 1), which can be modulated by remote influences (e.g. climate modes such as ENSO) and their teleconnections. The key processes, drivers, and teleconnections identified from our literature synthesis and confidence assessment, in addition to our composite analysis above, are summarized in Fig. 2. We now outline the characteristic processes, drivers, and teleconnections as critically assessed here across the four ocean-climate zones, and specifically for each case-study region.

**Tropical latitudes.** In the tropical Pacific, two-thirds of documented MHWs have been associated with El Niño events. The onset of El Niño coincides with trade wind weakening and reduced equatorial upwelling with the eastward propagation of warm water anomalies that deepen the thermocline along the equator[21]. The large increase in SST during El Niño in the eastern tropical Pacific near the Galápagos Islands and the west coast of South America can result in long duration MHW events in this region (Figs. 3 and 4c)[22].

The IOD, and ENSO via the IOB, are key climate modes contributing to the SST variability in the tropical Indian Ocean[23,24]. A positive IOD corresponds to anomalous warming in the western tropical Indian Ocean and anomalous cooling in the east that is generally active between austral winter and spring. It is notable that increased MHW occurrences in the tropical Atlantic are associated most strongly with a negative NAO while MHW occurrences in the tropical Indian Ocean most strongly

relate to a positive DMI (Fig. 3). The atmospheric processes associated with a negative IOD may have extended the duration of the intense MHW across northern Australia during the 2015/16 El Niño[25]. However, the co-occurrence of IOD and ENSO is more likely with a positive IOD during El Niño, and negative IOD during La Niña. IOB events are a direct response to ENSO and cause widespread warming of the Indian Ocean during and after the peak of El Niño, and cooling after La Niña[17,24,26]. We find that there is a secondary relationship between increased MHW occurrences with DMI evident in the central-eastern equatorial Pacific through its implicit correlation with ENSO[27].

In the case-study regions for tropical latitudes, we found that the longest-duration MHWs identified from the literature were around the Galápagos Island region during major El Niño events. The largest contiguous region experiencing MHW conditions occurred in early November 1997 (Table 2), while the record breaking 2015/16 El Niño was also linked to extreme SST conditions in the eastern equatorial Pacific. Mechanistically, the large SST changes in the central-eastern tropical Pacific during El Niño are due to coupled ocean-atmosphere processes (e.g. Bjerknes feedback) and changes in horizontal and vertical ocean heat transport. In the Great Barrier Reef region, El Niño conditions result in clearer skies, increased air-sea heat flux, weakened winds, warm ocean advection, and reduced vertical mixing, which are important drivers of MHWs. In the Seychelles region, the warmest SSTAs tend to be associated with ENSO, IOD, and ENSO-IOD events. The 1997/98 MHW event had co-occurring ENSO and IOD events[23], when easterly wind anomalies along the equator of the Indian Ocean weakened the trade winds to the south. The equatorial region warmed due to less latent heat release from the weakened winds, and eastward propagating equatorial Kelvin and westward propagating Rossby waves facilitated the basin-wide spread of the anomalous warming[28]. MHW events have been observed in the Andaman Sea of the Bay of Bengal in 1998, 2002, 2005, and 2010, with the April–May 2010 event being the strongest on record and causing severe coral bleaching[29]. Most of the MHWs in the Andaman Sea occurred during or after strong El Niño events in the Pacific, which often induced Indian Ocean basin warming and negative Indian Ocean Dipole after El Niño conditions. El Niño also induces warming events across the Bay of Bengal along the Indian and Sri Lanka coasts[30], likely associated with a weakened South Asian summer monsoon. The literature on Caribbean Sea warming events is limited to those reporting corresponding coral bleaching, where the causes and mechanisms of warming are not well documented or known, albeit that these reported events coincide with ENSO events[31,32].

In summary, ENSO dynamics and associated teleconnections are central to increasing MHW occurrences across tropical latitudes in the Pacific and Indian Oceans, with the IOD also being critical to occurrences in the Indian Ocean, but playing a secondary role in the Pacific. On longer timescales, the PDO/IPO modulates the frequency and intensity of El Niño events on decadal to multi-decadal time scales[8,33], which may be important for improving the predictability of MHWs in the tropics[33]. On synoptic and intraseasonal time scales, tropical MHWs can be generated through regional air-sea coupling via the Madden-Julian Oscillation (MJO) or atmospheric processes such as monsoonal variability[34]. The interactions between intraseasonal occurrences of MHWs and interannual-to-decadal climate variability have not yet been quantified.

**Middle and high latitudes.** Middle and high latitude ocean regions are considered poleward of 30 degrees latitude where the seasonal variability is much higher compared to the tropics.

MHWs here have been associated with shifts in warm ocean currents[35,36], mesoscale eddy activity, and atmosphere-ocean interactions[37–39]. MHWs have also been attributed to anomalous reductions in wind speed and wintertime surface cooling[13,37,39], shifts in the jet stream position[13,37,40] and the weakening of cold water currents and Ekman pumping[40]. Furthermore, ENSO teleconnections favour the seasonal persistence of large-scale weather patterns, such as high-pressure ridge systems that persist for multiple weeks, conducive to anomalous ocean warming[40,41].

Marine heatwaves in the Mediterranean Sea have been associated with large-scale atmospheric anomalies through increased heat flux and reduced wind speed that passively elevate SST well above their normal range[37,38,42,43]. Anomalous atmospheric circulation played a key role in the development of the 2012 northwest Atlantic marine heatwave[13]. The northward displacement of the jet stream can stabilize atmospheric blocking patterns of high pressure that reduce surface wind speeds and thereby inhibit ocean mixing, as well as increase surface air temperatures[44]. Large-scale atmospheric forcing can simultaneously warm large regions of surface water and cause large-scale MHWs. Blocking high pressure weather systems that develop over the Gulf of Alaska, in combination with remote tropical Pacific teleconnections, can cause sudden changes in oceanographic conditions leading to MHWs[41]. In the Bering Sea, large-scale atmospheric anomalies shift wind-driven oceanic frontal zones, increase the upper ocean stratification, and enhance heat flux into the ocean[45,46]. Reduced surface heat loss and sluggish circulation from weakened winds drive MHWs in the northeast Pacific[40,47–49]. Similarly, the South Pacific experiences MHWs from atmospheric pressure anomalies associated with ENSO that divert warm air poleward, weaken wind speeds, and slow the normal northward advection of cold water[50].

MHWs at the six middle and high latitude regions identified here all have a common driver. Large-scale atmospheric pressure anomalies precede anomalous ocean warming and are often triggered by remote changes in SST. Stalled ridges of atmospheric high-pressure systems cause clear skies, warm air, and reduced wind speeds. These conditions make rapid warming in the upper ocean possible, coupled with increase thermal stratification due to reduced vertical mixing. In the North Pacific, we identify multiple drivers that can enhance the occurrence of MHWs, including El Niño and the positive PDO and IPO phases (Figs. 3 and 4). El Niño mainly influences MHWs in the South Pacific case-study region (Figs. 3 and 4d). The northwest Atlantic MHWs are associated with the negative NAO phase (Fig. 4d), while the Mediterranean Sea is dominated by the positive DMI (Fig. 4d).

**Western boundary currents and extension regions.** WBCs are regions of poleward ocean heat transport at the western extent of the subtropical gyres, where intense air-sea interactions can also affect the local climate. They are associated with strong temperature fronts and eddy variability, with large variability in their transports and location across a range of timescales.

Recent studies suggest that the intensification and poleward displacement of WBCs around the globe are causing local and regional SST warming trends in the mid-latitudes that far exceed mean global warming rates[45,46]. Although the WBC regions and their extensions appear as prominent features when examining typical MHW intensity (Fig. 1b), there was a notable absence of reported events in the literature based on our search (Supplementary Table 1). Characteristically, WBC regions are prominent in MHW frequency and intensity (Fig. 1b, d) and do not show up particularly strongly in duration (Fig. 1c), reflecting the high SST variance in these regions due to relatively rapid transport

variations and large mesoscale eddy activity compared with other zones. WBCs are regions of exceptionally sharp horizontal SST gradients, such that small spatial displacements of the edge of the WBC can translate into very large SSTAs.

Given the lack of reported MHW events in WBC systems, it is not possible to generalize the local drivers that give rise to MHWs in this zone. Nevertheless, with their role in the poleward advection of heat, one might expect a persistent and enhanced poleward WBC transport to potentially result in a warm SST event (and possible MHW), although only one such case has been discussed in the literature[9] during the 2015/16 Tasman Sea MHW that occurred in the East Australian Current (EAC) Extension region located poleward of the EAC core separation point[9]. This event lasted for >8 months with an intensity unprecedented over the satellite record (i.e. since 1982). For a number of WBC regions, it has been noted that shifts in current location are associated with SSTAs[51–53], but these changes have not been directly linked to MHWs.

It has been shown that ENSO may weakly modulate western boundary current characteristics including SST on multi-year timescales via the influence of westward propagating oceanic Rossby waves[54–56] with the slow propagation of these waves precluding their signal being evidenced as concurrent teleconnections. In addition, Fig. 4b suggests that other interannual climate modes may also be important at zero lag together with the PDO. In contrast, the low frequency modulation of ENSO by the IPO/PDO, and thus the lagged dynamic response due to oceanic Rossby wave teleconnections, may have promise for the potential predictability of MHW likelihood on multi-year time scales in these WBC case-study regions. Nonetheless, most of these relationships are weak suggesting that a substantial fraction of MHWs in WBC systems may be internally generated. For example, off southeast Australia most of the interannual variance of the western boundary current system appears to be unrelated to large-scale climate modes[57] and instead may be due to the internal variability arising from shorter time scale instabilities generated from local forcing, interacting, and cascading across scales[58].

**Eastern boundary currents**. In most subtropical gyres, eastern boundary currents (EBCs) flow equatorward and are associated with the upwelling of cool, high nutrient water along the coast. The exceptions are the poleward flowing Leeuwin Current off Western Australia and the poleward flow off the northwestern Iberian Peninsula. MHW events in EBCs (Table 1, EBC) have been associated with anomalous poleward currents that advect heat and suppress upwelling. Tropical climate modes, especially ENSO, directly influence EBC systems due to the intersection of equatorial and coastal waveguides.

Off Western Australia, MHWs known as Ningaloo Niño[36] are associated with remote forcing from the tropical Pacific Ocean[14,18,59] as well as local factors including air-sea heat flux[15,60,61]. The Leeuwin Current system in the Indian Ocean is influenced by the tropical Pacific Ocean due to its oceanic connection via the Indonesian Seas[36,62]. The most well-known MHW occurred in 2011, an unprecedented event coinciding with the near-record La Niña event[36,60] owing to anomalously strong easterly winds in the equatorial Pacific, and caused by anomalous poleward advection from a record Leeuwin Current transport and anomalous air-sea heat flux into the ocean[3,36,60]. Interannual variability of the Leeuwin Current is predominantly driven by ENSO[63] and by local alongshore wind anomalies from atmospheric circulation anomalies[18,64]. The coupling between alongshore winds and coastal SST, as well as the mixed layer depth[15], is fundamental to the amplification of Ningaloo Niño events[61].

Since the late 1990s, Ningaloo Niño events have occurred more frequently associated with a cooling trend in the equatorial eastern Pacific (i.e. negative IPO phase) and enhanced ENSO variance[18], although some evidence points to a transition to a positive IPO in the last few years[65].

Off western Africa, MHWs have been documented in the Benguela Current system. These events have been linked with anomalous winds in the tropical Atlantic Ocean that give rise to the so-called Benguela Niño[66]. This remote forcing is associated with a relaxation of trade winds in the western tropical Atlantic Ocean and westerly wind anomalies along the equator[67–69]. Sea level height anomalies propagate toward Africa as equatorial Kelvin waves that subsequently travel southward along the coast. These conditions are associated with strong poleward geostrophic flow, thermocline deepening, and near-surface warming. Recently, the Dakar Niño, a coastal MHW off the west coast of North Africa, has become a new member of the coastal Niño phenomena, partly driven by coupled ocean-land-atmosphere interactions[70].

Off Baja California and Mexico, a coupled air-sea phenomenon known as the California Niño is associated with equatorward wind relaxation and anomalous oceanic advection causing warm SSTAs. Warm SSTAs lead to a heat flux out of the ocean and the atmospheric response maintains anomalous poleward surface currents, which give rise to a positive feedback[71]. During the extreme 1997/98 El Niño, this region was impacted by an anomalous large-scale cyclonic flow in the North Pacific subtropical gyre, which created an unusual pool of warm SSTAs by advection[72]. To the south, off the Peruvian coast (mid-west coast of South America), MHWs have been documented in association with El Niño, although these events were not specifically termed MHWs. These events arose from the high sea level anomalies propagating along the coast as coastal Kelvin waves, deepening the thermocline with contributions from weakening southeasterly trade winds[73,74]. These processes contribute to strengthening of poleward currents and advection of warmer water to the south.

In summary, with ENSO being central to tropical Pacific atmosphere-ocean dynamics, connections between equatorial and coastal Kelvin waves may be important to the onset of MHWs in the tropical eastern boundary regions of the Indian and Pacific Oceans. These relationships feature prominently in Fig. 4a, which reveals the tendency for a positive (negative) Niño3.4 to enhance MHW days off the west coasts of North and South America (west coast of Australia), while the negative phase of the EMI dominates the enhancement of MHWs off Peru and Western Australia[59]. However, it is important to recognize the regional complexities from air-sea coupling in these eastern boundary regions associated with regional Niño/Niña phenomena (e.g. Ningaloo and California). The positive phase of the PDO can contribute to enhancing MHW days off the west coast of North America. Off western Africa, the positive phase of the Atlantic Niño index is the dominant phase that enhances MHW days (Fig. 4a). During these phases, high sea surface height anomalies and a deepening of the thermocline result in anomalous poleward currents. Anomalous downwelling favourable winds can also contribute to the strengthening of these poleward flows. These currents result in the advection of warmer water poleward, regional air-sea heat fluxes, and local feedbacks that can contribute to localized warming.

## Discussion

Our assessment and analysis identified a number of local oceanic and atmospheric processes and large-scale climate phenomena that generate or modulate MHW events. Locally, air-sea heat

fluxes, horizontal and vertical advection of heat, and mixing processes play roles in generating, maintaining and terminating MHWs (Supplementary Fig. 1). For example, in boundary current and extension regions, horizontal advection may dominate (e.g. Ningaloo Niño[36] and the Tasman Sea MHW[9]), although these MHWs may also be strengthened or maintained via feedback processes involving air-sea interactions. In regions of strong temperature gradients (e.g. in some western boundary current extension regions; Fig. 1a), a shift in location of these fronts can also cause large increases in SST[75]. Modulation of tropical MHWs are well understood in the context of El Niño (or IOD) events. These events, which may be tied to a specific large-scale climate mode phase, are characterized as long duration MHWs in the equatorial Pacific Ocean caused by coupled atmosphere-ocean interactions including net downward surface heat fluxes and persistent changes in the winds that impact upwelling intensity, the depth of the thermocline, and horizontal advection, producing anomalously warm mixed layer temperatures and ocean-atmosphere feedbacks.

Ocean heating caused by air-sea heat flux anomalies can also be affected by both local and large-scale atmospheric conditions. For MHWs caused by surface heat flux anomalies, warmer SSTs are typically associated with atmospheric subsidence, clear skies, reduced wind stress, low surface level relative humidity, and/or warm surface air temperature. Changes in advection by ocean currents are largely related to changes in regional-scale, local, or remote surface winds. Weakening of the trade winds in the Pacific Ocean during El Niño result in a reduction of the westward flow of water, both locally and remotely because of eastward propagating equatorial Kelvin waves, allowing for a greater build-up of temperatures in the eastern equatorial Pacific. Also, the intensification of the trade winds during La Niña has been associated with high steric heights propagating through the Indonesian Seas and causing a remotely forced acceleration of the Leeuwin Current[63]. In contrast, the northeast Pacific MHW from 2013–2016 has been described as a continuous event involving various tropical-extratropical teleconnections in the spatial evolution and persistence of the SSTA field[8].

On multi-decadal timescales, anthropogenic ocean warming is likely to shift the probability distribution of MHWs. Using the same MHW definitional framework, Oliver et al.[11] show an increase by over 50% of globally-averaged MHW days over the last century, which can be largely explained by increases in mean SST. As ocean warming continues, some studies suggest that SST variability may also change. At regional scales, climate models project substantial intensification of transports in certain WBC extension regions[76,77], which would manifest as a change in eddy variability (e.g. ref. [78]). In recent decades, the extension of the EAC has already been linked with shifts in populations of marine species and consequent changes in the structure of communities and ecosystems in the Tasman Sea[79,80]. Events such as the unprecedented Tasman Sea MHW in 2015/16[9] may become more common. At basin scales, extreme El Niño, La Niña, and IOD events are projected to increase in frequency due to greenhouse warming[81–83]. These projected changes in climate modes may have substantial implications for MHW characteristics in the future. Given the importance of these modes in relation to MHWs, any such changes will likely have far-reaching impacts on marine ecosystems.

We used a common framework[6] to unify our understanding of historical MHW events across time and space. This quantitative approach enabled us to identify the statistical links between MHW occurrence and various modes of internal climate variability (Figs. 3 and 4). We found that most regions show significant changes to MHW occurrence during specific climate mode phases. In some case-study regions, the likelihood of MHWs was raised or lowered by close to a factor of two during particular phases. This raises questions around the potential predictability of MHW enhancement or suppression given knowledge around predictability of relevant climate modes (e.g. ref. [14]). The analysis also reveals that there may be compounding influences from multiple large-scale climate modes potentially affecting certain regions simultaneously, since many climate modes are not independent of each other.

Moving forward, a systematic procedure is needed to document MHWs and their ecological impacts as we consider projections of climate modes and the complexity of physical processes controlling upper ocean temperatures. Categorizing and/or naming MHWs is one approach[84]. Here we have highlighted our current understanding of major MHWs regionally and their links to climate drivers. We recommend that it would be beneficial for studies of MHW events to provide several levels of information, ranging from noted ecological impacts to the processes that may have been associated with the warming, and the active climate drivers linked with those processes. Our capacity to detect MHWs continues to improve with advances in remote sensing and greater spatio-temporal coverage by in situ instruments. Further, datasets are now becoming available that will allow us to understand the three-dimensional structure of MHWs. Ocean reanalysis products provide spatially and temporally continuous views of the subsurface ocean and ARGO profiles provide us with unprecedented subsurface coverage, although these have only been available since early this century. Strategic implementation of our monitoring capabilities will be of increasing importance into the future, in order to advance predictive capabilities of MHWs and provide rapid assessment to the community.

## Methods

**Systematic review.** *Web of Science* and *Google Scholar* were used to find relevant peer-reviewed literature from 1950 to February 2016 for our analysis. In our searches, we used the keywords: sea surface temperature, extreme event, temperature anomaly, heatwave (and heat wave), and anomaly. Of the potential case studies found, suitable studies for inclusion in our database analysis were identified following the application of qualitative selection criteria (Supplementary Table 3). As a result, a total of 84 refereed articles were included in the database (see Supplementary Tables 1 and 2). Given the rapidly growing literature on MHWs, a few important relevant papers were also included since the cut-off. We evaluated each study to identify the local to large-scale driver(s) of each of the reported extreme ocean temperature events.

**Confidence assessment.** For each case-study region (boxed in Fig. 1), we assigned a confidence rating to the drivers of the MHW events reported in the literature. To frame this confidence assessment, we adopted similar uncertainty protocols as those reported in the Intergovernmental Panel on Climate Change Fifth Assessment Report (IPCC AR5)[85] and undertook an expert confidence assessment of the literature following the logic in Supplementary Fig. 1. Confidence was ranked from very low, low, medium, high, and very high, and is a qualitative measure of the level of scientific agreement and amount of evidence that supports a particular finding (Supplementary Fig. 2). Here, we applied our expert interpretation of the agreement between, and amount of evidence underpinning, the case-study assessment (Table 1) of the relevant MHW drivers (listed and abbreviated in the Table 1 legend).

Traceable evidence to support findings and quantified levels of uncertainty was considered as strong agreement, and contingent upon both the quality and amount of evidence presented[86]. Sizeable evidence consisted of multiple (three or more) independent studies of high-quality evidence. A very high confidence rating corresponds to both sizeable evidence and strong agreement. The peer-review process should preclude unreliable evidence or conclusions. However, unavoidable scientific uncertainty is inherent in our own confidence measure. The adopted confidence ranking reconciles the diversity of analyses and data being used by considering overall agreement. For each MHW case-study, we have assessed and reported the level of confidence in the identified MHW driver component as being accurate, which is dependent on the level of agreement and amount of evidence. We consider this to be a systematic approach to comprehensively document MHWs, highlighting agreement and disagreement across the literature.

**Definition of MHW driver**. We define a MHW driver to be the set of causative mechanisms that combine to produce a MHW event. Specifically, a driver will comprise of the local processes that directly affect the anomalous ocean temperature tendency budget during a MHW event, and where relevant, combined with the large-scale climate forcing mechanisms (e.g. known climate modes, with a centre-of-action that might be remote to the MHW) via teleconnection processes. Our definitions for local processes, climate forcings, and teleconnection processes are given below.

Local processes (affecting the MHW budget): represent the processes that directly affect the evolution of ocean temperature (in particular, in the surface mixed layer) at a certain location, i.e. heat fluxes, horizontal advection (from the mean circulation or high-frequency small-scale flow), vertical entrainment, and horizontal and vertical mixing.

Climate forcings (modes of variability): represent the important large-scale climate modes of variability—like El Niño–Southern Oscillation (ENSO), the Indian Ocean Dipole (IOD), or the North Atlantic Oscillation (NAO)—that remotely or locally generate climate conditions that act to increase or decrease the occurrence of MHWs.

Teleconnection processes: represent the atmospheric or oceanographic processes (e.g. planetary waves, atmospheric bridge) that mechanistically connect the remote modulating climate mode forcing with the direct local process affecting the MHW temperature tendency budget.

The present study focuses on MHWs as expressed at the sea surface (we note that subsurface data are sparse and ecological impacts are generally realised in the upper ocean where biological productivity is greatest). Hence, local processes represent the surface mixed layer processes that govern extreme sea surface temperature anomalies (Supplementary Fig. 1) and that can be potentially diagnosed from a surface temperature (or heat) budget analysis. A list of the relevant mechanisms identified as important to MHW occurrence is provided in Table 1.

**Climate indices as metrics for climate modes**. In our composite analysis of MHWs, we considered nine commonly used climate indices that are associated with influential modes of variability in the climate system. These climate modes have characteristic time scales ranging from sub-seasonal (e.g. the SAM) to multi-decadal (e.g. the IPO). It is important to note, however, that there may be compounding influences of multiple large-scale climate modes potentially affecting certain regions simultaneously, since many climate modes are not independent of each other. The climate mode indices used are described below.

Niño-3.4 index—a measure of equatorial central Pacific SSTAs indicative of the state of ENSO (source: www.esrl.noaa.gov/psd/gcos_wgsp/Timeseries/Nino34/).

NAO index—consists of a north-south dipole in atmospheric pressure anomalies, with one center located over Greenland and the other center of opposite sign spanning the central latitudes of the North Atlantic between 35°N and 40°N. The index is based on a rotated principal component analysis of monthly standardized 500-mb height anomalies in the North Atlantic poleward of 20°N (source: www.cpc.ncep.noaa.gov/products/precip/CWlink/pna/nao.shtml).

PDO index—is defined as the leading principal component of North Pacific monthly sea surface temperature variability poleward of 20°N (source: research.jisao.washington.edu/pdo/).

TPI (Tripole index)—is used to indicate the state of the IPO, and is based on the difference between SSTAs averaged over the central equatorial Pacific and the average of SSTAs in the northwest and southwest Pacific (source: www.esrl.noaa.gov/psd/data/timeseries/IPOTPI/).

Atlantic Niño index (ATLN 1)—based on SSTAs in the tropical eastern South Atlantic (17–7°S and 8–15°E) "(refer https://rmets.onlinelibrary.wiley.com/doi/full/10.1002/asl2.424 and https://github.com/alexsengupta/MHWdata). A positive (negative) phase is associated with warm (cool) anomalies.

SAM index—measures the strength of the Southern Annular Mode and represents the surface pressure difference between 40°S and 65°S (source: www.antarctica.ac.uk/met/gjma/sam.html).

EMI—measures the strength of the Central Pacific ENSO (also known as El Niño Modoki)—the index is based on the difference between SSTAs in the central equatorial Pacific and the averaged eastern and western Pacific SSTA (source: http://www.jamstec.go.jp/frsgc/research/d1/iod/modoki_home.html.en).

DMI—measures the strength of the Indian Ocean Dipole represented by the anomalous SST difference between the western equatorial Indian Ocean (50°E–70°E and 10°S–10°N) and the south-eastern equatorial Indian Ocean (90°E–110°E and 10°S–0°N) (source: http://www.jamstec.go.jp/frsgc/research/d1/iod/e/iod/dipole_mode_index.html).

NPGO index—the North Pacific Gyre Oscillation index is based on the second mode of sea surface height variability in the northeast Pacific (source: www.o3d.org/npgo/npgo.php).

**Driver space and time scales**. Based on the literature review, we found that different drivers can cause MHWs over a range of space and time scales (Fig. 2). To distinguish between drivers acting on different scales, we categorized the physical drivers across several commonly classified time scales (i.e. synoptic, intraseasonal, interannual, decadal). We defined synoptic timescales as lasting from days to weeks. Intraseasonal refers to variability occurring on the order of 30–90 day cycles.

The most notable intraseasonal phenomenon in the tropics is the MJO. Interannual, or year-to-year, variability is associated with phenomena such as ENSO, Ningaloo Niño, Atlantic Niño, Benguela Niño and the IOD. Lastly, the decadal time scale includes phenomena such as the PDO (or IPO) or the AMO, which represent climate variations on the order of decades to multiple decades. Events are categorized according to which driver played the primary role in the associated region (Table 1).

**Links between MHWs and drivers**. We examined the relationships between key climate indices (Fig. 4), as simple metrics for the climate modes, and the likelihood of MHW occurrence in the area-averaged case-study regions (Fig. 1). This was done by summing the number of days at each location when the SSTA for a given region was classified as a MHW (based on[6] using a 90th percentile threshold) and determining whether the climate index was in a positive or negative phase based on the 1982–2016 daily NOAA OI SST V2 (AVHRR-only) dataset[87] (the MHW calculation uses a climatological baseline period of 1982–2012). Monte Carlo simulations were conducted to determine whether the number of days counted were greater or less than what might be expected by chance. For a given climate index, a synthetic time series was constructed with similar autocorrelation characteristics to the index (using a fourth order autoregressive model). We then re-summed the number of MHW days during positive and negative phases of the synthetic index. This process was repeated 10,000 times to produce a frequency distribution of the expected number of days. We used the 5th and 95th percentiles to form confidence intervals for the given mode and region.

To generate Fig. 3, a similar procedure was undertaken using 2° × 2° area-averaged SSTA, based on the linearly detrended anomaly field from the daily NOAA Optimum Interpolation Sea Surface Temperature version 2 (OISSTv2) dataset from 1982–2016, and linearly detrended climate indices. For each resulting time series, we identified which climate modes were significantly related (>90% level) to MHW occurrence using the Monte Carlo simulation. Figure 3 provides a spatial map of apparent climate mode influence. Here, the global patterns of identifiable climate modes and their respective phases are indicative of the dominant climate mode on both the enhancement (largest number of MHW days) and suppression (smallest number of MHW days) of MHW occurrences at each location.

**Characteristics of reported events from the MHW definition**. We have applied the MHW definition[6] in the analysis on the NOAA OI SST V2 data to evaluate and unify our understanding of the maximum intensity, duration, and spatial extent of MHW events identified from the literature (Table 2). Post-1982 events described in the literature were identified using the NOAA OI SST V2 dataset where they met the following criteria. Daily anomalies were manually examined to identify when a MHW event was first observed and when it terminated. In contrast to other analyses in this paper, for the identification of these spatially extended extreme MHW events, we identified regions and periods when SSTA >98th percentile (rather than the 90th percentile). A 90th percentile threshold resulted in too many small events that made it unclear when the main event was taking place. If SSTA for a given event dropped below the 98th percentile for <3 days, but then re-emerged, we considered this to be a continuing event (as long as regional SSTA remained warm, i.e. above the 98th percentile). If SSTA dropped below the 98th percentile in one region while building up to exceed the 98th percentile in a nearby region, we considered this to be the same event. In Table 2, for each historical event, we reported the dates and magnitudes when: the SSTA first exceeded and finally dropped below the 98th percentile threshold; reached the maximum SSTA (above climatological baseline); and, reached its maximum contiguous area with SSTA greater than the 98th percentile threshold.

## Data availability

We have used publicly available data only; new data were not generated as a result of this study. NOAA high resolution SST were provided by NOAA/OAR/ESRL PSD (Boulder, CO, USA) from their website (www.esrl.noaa.gov/psd/). Peer-reviewed publications assessed in this study are available from the relevant journals. URL addresses to the climate indices used are listed under 'Climate indices as metrics for climate modes' in the Methods.

## Code availability

The code used to analyze these data and generate the results presented in the study can be obtained from https://github.com/ecjoliver/MHW_Drivers. The search tools used, *Google Scholar* and *Web of Science*, are publicly available.

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

## Acknowledgements

This contribution is an outcome from the working group 'Marine Heatwaves—physical drivers and properties' (www.marineheatwaves.org) hosted at the University of Western Australia (UWA) Oceans Institute by T.W., D.A.S., N.J.H., and E.C.J.O. in January 2015. The working group received support for the workshop from a UWA Research Collaboration Award, a UWA School of Plant Biology synthesis grant, and the ARC Centre of Excellence for Climate System Science (ARCCSS: CE110001028). This work contributes to the World Climate Research Program (WCRP) Grand Challenge on Extremes. This paper makes a contribution to the interests and activities of the International Commission on Climate of IAMAS/IUGG and National Environmental Science Programme (NESP) Earth Systems and Climate Change Hub (ESCC) Hub Project 2.3 (Grant No. B0024391). N.J.H. and L.V.A. acknowledge support from the ARC Centre of Excellence for Climate Extremes (Grant No. CE170100023), S.E.P. was supported by ARC grant DE140100952, M.G.D. by ARC grant DE150100456, D.S. by NERC grant IRF NE/K008439/1, T.W. by ARC grant FT110100174 and DP170100023. M.T.B. by NERC grant NE/J024082/1, J.B. acknowledges support from CE110001028, E.C.J.O. by ARC grant FS110200029, P.J.M. by Marie Curie CIG PCIG10-GA-2011–303685 and NERC grant NE/J024082/1, S.C.S. by an Australian Government RTP scholarship. Thanks to Dr Axel Durand for assistance with the Mendeley database. The daily 0.25° resolution and the weekly 1° resolution NOAA OI SST V2 data are provided by the NOAA/OAR/ESRLPSD, Boulder, Colorado, USA, at http://www.esrl.noaa.gov/psd/. Finally, we would like to thank the three reviewers (Nathan Mantua and two anonymous) for their detailed and constructive comments, which led to significant improvements in the final manuscript.

## Author contributions

N.J.H. formulated the initial concept, led the research activity, and coordinated the writing. N.J.H., H.A.S., J.A.B., M.F. and A.S.G. led the overall project design, and all other authors (E.C.J.O., L.V.A., M.T.B., M.G.D., A.J.H., P.J.M., S.E.P.-K., D.A.S., S.C.S. and T.W.) contributed to further developing the concept during the workshop at UWA. H.A.S. led and undertook the literature analysis. A.S.G. designed and performed the statistical analysis between climate modes and MHWs. A.S.G., H.A.S., J.A.B. and E.C.J.O. generated the figures. All authors (N.J.H., H.A.S., A.S.G., J.A.B., M.F., E.C.J.O., L.V.A., M.T.B., M.G.D., A.J.H., P.J.M., S.E.P.-K., D.A.S., S.C.S. and T.W.) discussed the results, aided in their interpretation and contributed to the writing.

## Additional information

**Competing interests:** The authors declare no competing interests.

