## [Peer Review File · Nature Communications]

Reviewer #1 (Remarks to the Author):

Holbrook Nature 2018 review

The approach taken here is to combine a literature review and synthesis with a criteria for identifying MHW events (SSTA > 98th percentile values for at least 5 consecutive days), and then correlating MHW statistics with large-scale climate indices. The analysis includes two tables and four figures, and an extended Supplemental Information section that provides more details of the MHW literature review that is synthesized in the main text. I found this manuscript very difficult to review. For me, it fails to present a coherent structure for synthesis and interpretation of the MHW literature, and fails to deliver a clear set of key findings that will help readers come to some new understanding for the main drivers of recent MHW events in different parts of the world, and how those main drivers vary in different types of oceanographic settings.

All that said, I think this work has the potential to deliver a coherent and interesting synthesis and interpretation, but to do this will require a major revision of this manuscript. The basic elements are here, but need to be better organized into a simpler framework, and then better explained and supported with the literature review. Elements of this framework are already in the main text in lines 46-49:

MHWs are the direct result of local-scale processes which can be modulated by remote influences (like ENSO) and their teleconnections.

Next I would offer a brief review of local processes that impact the upper ocean's heat budget and SST. Then I would take pains to expand upon this relatively simple foundation. Specifically, my read of the various drivers identified in different MHW events in the SI section basically comes down to 3 classes of drivers: (1) unusually persistent local atmospheric forcing, (2) unusually persistent remote atmospheric forcing, or (3) instabilities in upper ocean current systems that may be influenced by (1) and (2), or may be intrinsic to the ocean circulation. Unusually persistent local and remote atmospheric forcing can arise from either intrinsic atmospheric variability (like the NAO, or blocking events, etc.), or coupled ocean-atmosphere modes of variability (like ENSO or the IOD) and their tropical/extratropical teleconnections.

Then from this broadening foundation, one could expand upon the actual processes of heat exchange associated with the different categories of drivers

- anomalously persistent local atmospheric forcing can include air-sea heat fluxes, horizontal advection, vertical mixing and thermocline changes, etc.
- remote atmospheric forcing can generate propagating internal waves and upper ocean currents that then alter the local energy budget of the upper ocean through advection, currents, or thermocline displacements

- instabilities in upper ocean current systems can alter the mean flow or eddy transports

Basically, the text needs to do a better job describing the ways in which the various items identified in Figure 2 are related to MHW events in different oceanographic settings, and how these relationships have played out in MHW's in the historical record. It should lead to a coherent explanation for the main drivers of MHW events in EBCs, WBCs, the tropics, and mid-to-high latitudes if those main drivers are a consequence of the dynamic and thermodynamic setting of the different types of ocean systems.

To make this manuscript even more complete and compelling, it would be great to include some kind of cluster analysis or PCA that used what looks to me to be a very messy data set in Table 1 to more objectively demonstrate that MHW events in the different system types do, indeed, tend to involve the same set of main drivers.

If the authors can pull this off, it has the potential to make for a publication worthy of being published in Nature, and I hope they can do it.

In contrast, the analysis presented in this manuscript attempts to link different MHWs to large scale modes of climate variability through correlations between gridded SST fields and various climate indices. I don't find this approach to be very informative given the fact that many of the large scale climate indices considered are developed from (or are highly correlated with) specific geographic patterns of SST variations. I just don't find it to be informative that ENSO is a driver of MHW events in the eastern equatorial Pacific when that is the oceanic signature of ENSO, or that the AMO is a driver for North Atlantic MHWs when the AMO index is based on North Atlantic SSTs, etc. The fact that clear patterns emerged relating climate modes and MHWs across the globe does not seem to be a novel or interesting result to me, and it would be extremely surprising if such patterns didn't exist given the way climate indices are developed to account for recurring patterns of climate variation that capture substantial fractions of variance in the global climate system.

In the analysis presented here, perhaps the most interesting finding is that MHWs around the western boundary currents are not well-correlated with the large-scale climate indices considered in this analysis. The discussion of this finding should be explored in the context of the different types of drivers, and how they are connected with (or influenced by) the large-scale modes of variability.

I have a number of additional specific comments for the authors to consider below.

line 21: Is there really a lack of understanding around the physical processes that give rise to MHWs? I think this statement is at odds with the sentence on Lines 46-49 stating "MHWs are the direct result of local-scale processes (e.g. advection, air-sea interaction or vertical mixing) which can be modulated by remote influences and their teleconnections".

line 25-26: wasn't a similar quantitative framework used by Hobday et al. (2018, Oceanography) to uniformly and systematically rate the characteristics of MHWs from satellite (NOAA OISST v2) data? What is new here, adding measures for area extent and changing the criteria from 90th percentile to 98th percentile such that the events are more intense, shorter in duration, and cover smaller areas than those previously identified?

line 39: here it says that the longest duration events occur in the E. Eq. Pacific associated with El Niño - yet using criteria of Hobday et al. (2018, Oceanography) the 2014-2016 MHW in the NE Pacific falls out as an event that persisted for over 700 days.

lines 63-64: I find it hard to distinguish between a strong El Niño and a MHW in the ETP. Saying that "the arrival of El Niño pre-conditions the ETP ..." is like saying "an extended spell of extreme hot weather pre-conditions a region for a heat wave", isn't it?

line 80: with respect to drivers of MHW events in the Indian Ocean - IOD is a dominant mode of SST and wind stress variability in the Indian Ocean, but itself is not a driver. The drivers are the changes in surface wind stress and related air-sea heat fluxes, Ekman transports, and thermocline displacements, and the feedbacks between the IOD pattern of SST and surface wind stress, aren't they?

lines 82-84: again, saying that the IOD caused the extended duration of the MHW doesn't make sense. The IOD is a mode of coupled ocean-atmosphere climate variation that involves various heat exchange mechanisms and feedbacks between the ocean and atmosphere that promote persistence of the event and its spatial pattern, so it makes more sense to me to describe the events in this way.

line 100: I would replace "cause" with "favor the seasonal persistence of" ... ENSO related teleconnections are typically defined using monthly or seasonal averages, while "weather" is typically defined with much higher-frequency data (hourly-daily)

line 104: "decreased input of cold Arctic water ..." this is not correct - Arctic water does not influence SSTs in the Gulf of Alaska, and was not a factor in the NE Pacific warm blob in 2013-2016. Anomalously weak cooling of the warm blob region in fall 2013/winter 2014 was due to a combination of anomalously weak vertical mixing, Ekman transports, and surface heat fluxes from the ocean to the atmosphere, all related to the persistence of anomalously high atmospheric pressure that caused anomalously weak winds, clear skies and reduced storminess. The persistence and evolution of the local/regional atmospheric forcing has been the subject of several journal articles, and the nature of the local atmospheric forcing varied over the course of the 2013-2016 period (see your reference 22).

lines 128-134: this is the kind of description that makes a lot of physical sense to me, wherein the local/regional ocean-atmosphere heat exchange processes are described, and the larger scale context is provided.

line 147: insert "local/regional" between "causing SST"

lines 153-154: WBC's are also regions of exceptionally sharp horizontal SST gradients, such that small spatial displacements of the edge of the WBC translate into very large SST anomalies.

lines 165-166: do anomalous poleward currents in EBCs suppress upwelling? Is this specific to remotely forced alongshore currents? Or are there regional wind anomalies contributing to anomalous poleward currents, weaker upwelling and reduced heat flux from the ocean to the atmosphere all at the same time?

line 203: delete "drivers" from the end of this heading

line 210: replace "can suppress or enhance" with "are associated with enhanced or suppressed likelihood"

line 214: replace "effect" with "association with"

line 216: replace "influence on" with "association with"

lines 243-245: consider adding a new set of figures showing the regions having MHW statistics associated with each of the climate modes considered in this study to provide readers a cleaner/clearer view of their associations and potential influences, or don't show Fig. 4 at all.

line 251: end this sentence by adding "in the NE Pacific"

line 255-256: Would be best to note that statistical relationships do not necessarily indicate causal links much earlier in this article

lines 262-263: the lack of correlations with climate modes may simply indicate that persistent changes in local or remote forcing (e.g. wind stress) that is not associated with a large scale mode of variability is the key driver

lines 276-278: ENSO physics is better described as involving coupled ocean/atmosphere interactions that include changes in winds that impact upwelling intensity, the depth of the thermocline, horizontal advection, o/a heat fluxes and SSTs, and SSTs that impact atmospheric sea level pressure, surface winds, and convection.

lines 288-291: Di Lorenzo and Mantua (2016) described the NE Pacific MHW as a continuous event that involved various tropical-extratropical teleconnections in the spatial evolution and persistence of the SST anomaly field, but not as an event that faded then re-emerged.

line 304: with regard to “modes as drivers of MHWs ...” - some modes may be drivers, but others are defined by SST patterns so are diagnostics rather than drivers

line 331: what was the cutoff date for your web of science/google scholar search?

Figure 4: too much information - I find these maps to be extremely difficult to interpret, perhaps you could make individual maps for each climate model considered in Supplemental Material

Table 1: too much information - I find the information presented here to be extremely difficult to interpret

Table 2: I don't think it is meaningful to have 4 distinct MHW events in the Humboldt Current associated with both the 1982/83 and 1997/98 El Niño events, and 3 distinct California MHW events associated with the 1997/98 El Niño. The dynamics of ENSO are such that the basin-scale/global-scale tropical event has a duration of ~6 to 18 months, with peak amplitude and maximum teleconnection to the NE Pacific in the boreal winter months. Each of those sub-seasonal events is really part of the same ENSO event, and if you look at SST anomaly time series for the MHW regions of interest you will not see an end to the warm anomalies, just temporary dips below the 98th percentile

Table 2: you could add the 2014-2016 record warming of the CCS for “California” (See Jacox et al. 2018, Bull. of the Am. Met. Soc.)

Table 2: for the Northeast Pacific Ocean, 2013-2016 had two separate events, and not one? This seems at odds with the evolution and persistence of extreme SSTA in that period and how it has been reported in other publications (DiLorenzo and Mantua 2016; Jacox et al. 2018)

Nate Mantua

NOAA/NMFS/SWFSC

Santa Cruz, CA

Reviewer #2 (Remarks to the Author):

see attached PDF

Reviewer #3 (Remarks to the Author):

This is a literature review and synthesis of Marine Heat Waves globally. The manuscript summarizes the characteristics of heat waves appearing around the world since 1982, and a global SST data set is used to then relate heat waves to known modes of ocean-atmosphere variability. There is a table that summarizes findings from the literature geographically, and also a figure that relates phases of different modes with temperature increases or decreases. In addition, a figure which schematically places heat waves into the context of time and space scales of ocean variability is included.

The scope of this work and the synthesis is sound, but is somewhat limited because only sea surface temperature is analyzed. Since this is primarily a review of previous findings, with further analysis of SST data, there is not a comprehensive heat budget presented, or consideration of the vertical scale of the temperature anomalies. There is a statement (lines 375-376) that sub-surface data are sparse, and that ecological impacts are generally realized in the upper ocean. While the SST analysis is enlightening, a more thorough treatment would include a full heat budget analysis of the surface mixed layer. Since the subject is of global extent, this is impossible with the lack of sub-surface observations. However, given the wide range of space and time scales in Figure 2, there needs to be some discussion of how the SST analysis relates to changes in the mixed layer or upper ocean. Some of the processes that are included in Figure 2, such as ENSO or Kelvin waves, do have known signals in terms of changes in the depth of the mixed layer or changes in stratification. Ecosystem effects are concentrated in the upper ocean, but mixed layer properties are likely more indicative than SST. Some discussion is needed of how the heat waves might be affecting mixed layers. This can be a general statement, but the consideration of only SST is a big limitation and needs a more general context for upper ocean changes.

There needs to be a statement in the main text that the SST analysis includes only events within the 98th percentile or above. The common usage before has been 90th percentile. I did not see this until the very end of the supplemental material. This needs to be featured in the text.

Figure 4 is practically unintelligible. The intent of the figure and the analysis that went into it is admirable, but between the large number of modes/colors, the extreme pixel to pixel variations, and the small size of the figure, it was very difficult to interpret. Just looking at the north Atlantic for example, it is hard to see any signal in the central part of the basin away from the NAO dominant regions in the north and south. Don't know if this needs to be split into more panels (eastern and western hemisphere) but it is just too much information.

On the whole, the analysis is sound, the geographical coverage is extensive, and the attempt to establish dominant drivers is novel and important. Providing a better context for the results presented by having some discussion of vertical scales/mixed layer variability would greatly improve the impact of the analysis and synthesis. Efforts such as the ARGO floats have provided a lot of global scale information on temperature fields, and so future efforts to study heat waves on large space and time scales will need to include subsurface information in addition to SST.

The results are significant, and with the above issues addressed this would be worthy of publication.

Reviewer #1

Thank you for the thoughtful and constructive review of our manuscript. We provide our point-by-point responses (in black) to the comments and issues raised (in blue) and have revised our manuscript accordingly. In our responses, please note that the identified line numbers refer to our clean version of the revised manuscript, rather than the track-change version (where the track-changes are extensive).

The approach taken here is to combine a literature review and synthesis with a criteria for identifying MHW events (SSTA>98th percentile values for at least 5 consecutive days), and then correlating MHW statistics with large-scale climate indices. The analysis includes two tables and four figures, and an extended Supplemental Information section that provides more details of the MHW literature review that is synthesized in the main text. I found this manuscript very difficult to review. For me, it fails to present a coherent structure for synthesis and interpretation of the MHW literature, and fails to deliver a clear set of key findings that will help readers come to some new understanding for the main drivers of recent MHW events in different parts of the world, and how those main drivers vary in different types of oceanographic settings.

All that said, I think this work has the potential to deliver a coherent and interesting synthesis and interpretation, but to do this will require a major revision of this manuscript. The basic elements are here, but need to be better organized into a simpler framework, and then better explained and supported with the literature review.

Response: We acknowledge the reviewer's difficulty with reviewing our manuscript (we apologise) and have now restructured the information and largely rewritten the manuscript so that it is hopefully much clearer, coherent and informative. As recommended by the reviewer, we now present the analytical framework in a simpler (more straightforward) manner, outlined up-front in our revised manuscript introduction, where we state:

"The focus of this study was to develop and apply a consistent framework for understanding how regional MHW events relate to different modes of climate variability and the physical conditions causing waters to warm above a threshold. This framework aimed to inform the predictability of MHW events and, on a global scale, our capability to detect and understand how MHWs emerge. A useful lens for understanding the formation, maintenance and decay of MHWs is the upper ocean mixed layer heat budget, which includes the local processes responsible for changes in surface ocean temperatures. For MHW events, the dominant contributions to the temperature tendency within the mixed layer¹³⁻¹⁵ are given by

$$\frac{\partial T}{\partial t} = -\frac{1}{H} \int_{-H}^0 (\mathbf{u} \cdot \nabla_h T) dz + \frac{Q}{\rho C_p H} + \text{residual} \quad (1)$$

where the first term on the right hand side (RHS) is the horizontal advection (ocean advective fluxes) owing to the depth-dependent horizontal velocity vector \mathbf{u} and temperature T , the second term is the contribution for the net air-sea heat flux Q , where ρ is the average seawater density, C_p is the specific heat capacity of seawater ($4000 \text{ J kg}^{-3} \text{ }^\circ\text{C}^{-1}$), H is the mixed layer depth, and the residual third term includes horizontal eddy heat fluxes and the heat flux at the bottom of the mixed layer owing to vertical diffusion, entrainment, and vertical advection. To generate MHWs, the physical processes on the RHS of (1) have to contribute a sufficiently large net positive temperature tendency to increase the surface ocean temperature over a threshold⁶. Oceanic advective fluxes include unusually intense poleward flow, associated with the transport of warmer waters, enhanced heat advection by

boundary currents or reduced turbulent mixing. MHW favourable air-sea heat fluxes include contributions from anomalous net downward surface radiation, with little cloud cover and convective air-sea heat fluxes, and with suppressed latent heat loss from anomalously weak surface winds. While Equation (1) explicitly describes temperature variations that extend through the mixed layer, MHWs may extend deeper. The intensity of surface intensified MHWs may also depend on the state of the subsurface ocean, such as shallowing of the mixed layer depth^{10,15}.

In the vast majority of research conducted to date, and in our global analysis presented here, MHWs have been identified and characterised based on sea surface temperature (SST). This focus reflects the relative scarcity of data from the subsurface ocean, and the reported ecological impacts that are largely prevalent in the upper ocean where biological productivity is greatest. Key mechanisms that drive temperature changes in the mixed layer may include local internal variability (e.g. eddy instabilities) or large-scale modes of climate variability that act to modulate the local conditions – either from (1) local sources, for example, extreme ocean-atmosphere coupled feedbacks in the eastern equatorial Pacific during extreme El Niño – Southern Oscillation (ENSO) events, or (2) remote sources via ‘teleconnection’ mechanisms by, for example, the propagation of planetary waves in the atmosphere (that give rise to distant changes in the surface winds, cloud cover, etc.) or ocean (that can change the depth of the thermocline and drive remote circulation changes).

Our analytical framework was as follows. Our first goal was to identify localities and regions where robust statistically significant relationships exist between surface signatures of MHW occurrences and large-scale climate modes. We recognise and caution that significant statistical relationships do not necessarily indicate causal links. However, these analyses are indicative of potentially important connections that can provide clear pointers to potential predictability. Importantly, the relationships we have found and will highlight in the results between large-scale climate modes and MHW occurrences do not simply mirror those between climate modes and SST anomalies more generally.

Our second goal was to perform a global synthesis of the existing peer-reviewed literature on historical MHWs since 1950 (summarised in the supplementary online material (SOM)) that have occurred in oceanographically and/or climatically distinct regions (Fig. 1). We then conducted a confidence assessment (see Methods) of their characteristic drivers and processes (summarised in Table 1) to determine:

- a) The processes in the mixed layer temperature budget (RHS of Equation 1) important for MHW formation;
- b) Climatological systems and their configurations (e.g. high/low pressure systems, jet stream location), and teleconnection processes (e.g. atmospheric or oceanic Kelvin and Rossby waves), that either force or modulate the temperature budget terms; and
- c) Large-scale modes of climate variability that have played a role in modulating individual identified MHWs, either locally or via teleconnection.

From our synthesis and confidence assessment of the literature, the dominant drivers and processes were summarised as occurring on daily to decadal time scales and over sub-mesoscale to basin scale features (Fig. 2). By framing our global assessment into 22 case study regions (Table 1), we have generated, for the first time, a baseline regionalisation and globally assessed view of current knowledge of the characteristic drivers and processes of MHWs against which objective criteria for future quantitative analyses can be applied.

Our third goal was to revisit and further analyse these reported events since 1982 by producing consistent measurement estimates of the key MHW characteristics based on daily satellite SST data (Table 2). These characteristics are estimated based on a consistent MHW definition⁶ which allows for comparability (note a simple rescaling was uniformly applied (see Methods) to identify strong MHW events) given that the MHWs identified in our literature search have been reported using a diverse set of criteria.”

Elements of this framework are already in the main text in lines 46-49: MHWs are the direct result of local-scale processes which can be modulated by remote influences (like ENSO) and their teleconnections.

Response: In our submitted manuscript LL46-49, we wrote: “MHWs are the direct result of local-scale processes (e.g. ocean heat advection, air-sea interaction or vertical mixing; Fig. S1) which can be modulated by remote influences (e.g. climate modes, such as the El Niño-Southern Oscillation (ENSO)) and their teleconnections (Fig. 2).”

Our revised text is more direct and included at the top of the Section entitled **Assessment of the literature on drivers and processes** “Based on our global confidence assessment of the historical peer-reviewed literature on the local to large-scale mechanisms that cause MHWs, we have collectively determined that surface MHWs are the direct result of local-scale processes acting within the mixed layer (e.g. ocean heat advection, air-sea interaction or vertical mixing; Fig. S1) which can be modulated by remote influences (e.g. climate modes, such as ENSO) and their teleconnections. The key processes, drivers and teleconnections identified from our literature synthesis and confidence assessment, and extended by our composite analysis above, are summarised in Fig. 2.”

Next I would offer a brief review of local processes that impact the upper ocean’s heat budget and SST. Then I would take pains to expand upon this relatively simple foundation. Specifically, my read of the various drivers identified in different MHW events in the SI section basically comes down to 3 classes of drivers: (1) unusually persistent local atmospheric forcing, (2) unusually persistent remote atmospheric forcing, or (3) instabilities in upper ocean current systems that may be influenced by (1) and (2), or may be intrinsic to the ocean circulation. Unusually persistent local and remote atmospheric forcing can arise from either intrinsic atmospheric variability (like the NAO, or blocking events, etc.), or coupled ocean-atmosphere modes of variability (like ENSO or the IOD) and their tropical/extratropical teleconnections.

Then from this broadening foundation, one could expand upon the actual processes of heat exchange associated with the different categories of drivers

- anomalously persistent local atmospheric forcing can include air-sea heat fluxes, horizontal advection, vertical mixing and thermocline changes, etc.
- remote atmospheric forcing can generate propagating internal waves and upper ocean currents that then alter the local energy budget of the upper ocean through advection, currents, or thermocline displacements
- instabilities in upper ocean current systems can alter the mean flow or eddy transports

Response: We appreciate the recommended structural approach outlined here by the reviewer and have quite closely followed this approach outlining the framework in our revised introduction. Specifically, our manuscript has been restructured based on the organising principle that a marine heatwave (MHW) occurrence (identified here as a sea surface temperature anomaly expressed as a persistent temperature extreme) is underpinned by a local mixed layer temperature budget – see Equation (1) – while the local- and large-scale processes in Fig. 2 now correspond more directly to those presented in Table 1. With this restructure, the presentation is more logical and meaningful. As a consequence, we believe the findings should now be clearer and more informative, and the new knowledge better communicated. In redrafting Fig. 2 the terminology is now internally consistent with Table 1.

Basically, the text needs to do a better job describing the ways in which the various items identified in Figure 2 are related to MHW events in different oceanographic settings, and how these

relationships have played out in MHW's in the historical record. It should lead to a coherent explanation for the main drivers of MHW events in EBCs, WBCs, the tropics, and mid-to-high latitudes if those main drivers are a consequence of the dynamic and thermodynamic setting of the different types of ocean systems.

Response: Our Results section has been largely rewritten to provide a clearer assessment for each case study region. In our Results, we now more clearly outline the characteristic drivers (local processes, large-scale climate modes and teleconnections) as critically assessed across the four ocean climate zones, and specifically for each case study region.

To make this manuscript even more complete and compelling, it would be great to include some kind of cluster analysis or PCA that used what looks to me to be a very messy data set in Table 1 to more objectively demonstrate that MHW events in the different system types do, indeed, tend to involve the same set of main drivers.

Response: As outlined above, we have substantially revised the manuscript text and redrafted Fig. 2 so that the terminology is now internally consistent with Table 1, and hence the messages are more informative. Rather than apply a cluster analysis or principal components analysis to published papers, which do not confirm physical links due to the linear statistical nature of the analyses, we instead reinforce the commitment to, and undertaking of, our confidence assessment of the drivers identified in these peer-reviewed published papers. By applying expert knowledge in a confidence assessment framework, we provide levels of confidence for these drivers and processes at the selected space and time scales in Table 1 that we know are physically meaningful.

If the authors can pull this off, it has the potential to make for a publication worthy of being published in Nature, and I hope they can do it.

Response: Thank you for your encouragement. We hope that with the comprehensive rewrite of our manuscript, together with the revised Fig. 3 and new Fig. 4, that our revised manuscript is now much clearer in its presentation, framework, findings and contribution.

In contrast, the analysis presented in this manuscript attempts to link different MHWs to large scale modes of climate variability through correlations between gridded SST fields and various climate indices. I don't find this approach to be very informative given the fact that many of the large scale climate indices considered are developed from (or are highly correlated with) specific geographic patterns of SST variations. I just don't find it to be informative that ENSO is a driver of MHW events in the eastern equatorial Pacific when that is the oceanic signature of ENSO, or that the AMO is a driver for North Atlantic MHWs when the AMO index is based on North Atlantic SSTs, etc. The fact that clear patterns emerged relating climate modes and MHWs across the globe does not seem to be a novel or interesting result to me, and it would be extremely surprising if such patterns didn't exist given the way climate indices are developed to account for recurring patterns of climate variation that capture substantial fractions of variance in the global climate system.

Response: We acknowledge that sea surface temperature anomalies (SSTA) and various indices of modes of climate variability can be implicitly related. In the case of MHWs, however, we find important differences and substantive additional value to our understanding of the drivers of these events from our analysis.

To more clearly demonstrate the importance of climate modes to MHWs, and the relevant contribution, we have deconstructed our composite analysis further. Specifically, we have now examined the relationships between individual modes of climate variability and, separately,

regressed on SSTA and with a composite analysis on MHW occurrences. Here, the percentages are relative to the median number of MHW days (shown in Fig. 3 and the SOM), e.g. in a region where we expect MHWs to occur 10% of the time, an 80% increase means we would expect MHWs to occur 18% of the time associated with that phase of the associated mode. Further, the contributions from all significant modes to MHW enhancement/suppression for each case study region are summarised in Fig. 4. Fig. 3a shows for example that in the equatorial and far North Atlantic Ocean, and the tropical southwest Pacific (Coral Sea), there are likely to be >40% more MHW days during the NAO negative phase (see SOM Figs. S3-S15 for all results).

We believe our results from the deconstructed analysis should be more illuminating of the contribution from the analysis. Importantly, we find that SSTA and MHW occurrences are not characterised in the same way in relation to these climate modes. Rather, we find clear asymmetries (and regional differences) in the spatial distributions of percentage increase or decrease in MHW occurrences across the globe that are very different from those distributions found from climate mode regressions on +/- SSTA alone. On Lines 145-149, we note *“The relationships between MHW occurrences and climate modes (Fig. 3) are complex, although the large-scale patterns are broadly consistent with known SST anomaly patterns (see SOM – however there are also clear differences, e.g. compare the regressed SSTA against the composite MHW occurrence plots in Figs. S11 and S14 (SOM) for the North Atlantic Oscillation (NAO) and Southern Annular Mode (SAM) as climate mode drivers)”*.

Further, these figures effectively characterise the potential for predictability of MHWs associated with climate modes. That is, given a perfect forecast, these figures provide spatial maps of the distribution of percentage likelihoods of MHW increased or decreased occurrence in relation to each phase of the individual climate modes.

In the analysis presented here, perhaps the most interesting finding is that MHWs around the western boundary currents are not well-correlated with the large-scale climate indices considered in this analysis. The discussion of this finding should be explored in the context of the different types of drivers, and how they are connected with (or influenced by) the large-scale modes of variability.

Response: We have revised the **Western Boundary Current and Extension** sub-section of the **Results** section. Specifically, we now include:

“Recent studies suggest that the intensification and poleward displacement of western boundary currents around the globe are causing local/regional SST warming trends in the mid-latitudes that far exceed global average rates^{44,45}. Although the WBC regions and their extensions appear as prominent features when examining typical MHW intensity (Fig. 1b), there was a notable absence of reported events in the literature based on our search (Table S1). Characteristically, WBC regions are prominent in MHW frequency and intensity (Fig. 1b,d) and do not show up particularly strongly in duration (Fig. 1c), reflecting the high SST variance in these regions due to relatively rapid transport variations and large mesoscale eddy activity compared with other zones. WBCs are regions of exceptionally sharp horizontal SST gradients, such that small spatial displacements of the edge of the WBC can translate into very large SST anomalies.

Given the lack of reported MHW events in WBC systems, it is not possible to generalise the local drivers that give rise to MHWs in this zone. Nevertheless, with their role in the poleward advection of heat, one might expect a persistent and enhanced poleward WBC transport to potentially result in a warm SST event (and possible MHW), although only one such case has been discussed in the literature⁹ in a WBC extension: the 2015/16 Tasman Sea MHW that occurred in the East Australian Current (EAC) Extension region located poleward of the EAC core separation point⁹. This event lasted for >8 months with an intensity unprecedented over the satellite record (i.e. since 1982). For a

number of WBC regions, it has been noted that shifts in current location are associated with SST anomalies⁵⁰⁻⁵², but these changes have not been directly linked to MHWs.

It has been shown that ENSO may weakly modulate western boundary characteristics including SST, on multi-year timescales, via the influence of westward propagating oceanic Rossby waves⁵³⁻⁵⁵ - with the slow propagation of these waves precluding their signal being evidenced as concurrent teleconnections. In addition, Fig. 4b suggests that other interannual climate modes may also be important at zero lag together with the PDO. In contrast, the low frequency modulation of ENSO by the IPO/PDO, and thus the lagged dynamic response due to oceanic Rossby wave teleconnections, may have promise for the potential predictability of MHW likelihood on multi-year time scales in these WBC case study regions. Nonetheless, most of these relationships are weak suggesting that a substantial fraction of MHWs in WBC systems may be internally generated. For example, off southeast Australia most of the interannual variance of the western boundary current system appears to be unrelated to large - scale climate modes⁵⁶ and instead may be due to the internal variability arising from shorter time scale instabilities generated from local forcing, interacting and cascading across scales⁵⁷.

I have a number of additional specific comments for the authors to consider below.

line 21: Is there really a lack of understanding around the physical processes that give rise to MHWs? I think this statement is at odds with the sentence on Lines 46-49 stating “MHWs are the direct result of local-scale processes (e.g. advection, air-sea interaction or vertical mixing) which can be modulated by remote influences and their teleconnections”.

Response: We believe it is appropriate to say that “scientific understanding of marine heatwaves is in its infancy compared to that of atmospheric heatwaves”. We have modified the words in the sentence following to read “*In particular, there is limited understanding of the processes that give rise to MHWs^{7,8} and their characteristics, and ...*”. The apparent conflict with the statement on Lines 46-49 of the submitted manuscript has also been addressed by revising the statement as “*Based on our global confidence assessment of the historical peer-reviewed literature on the local to large-scale mechanisms that cause MHWs, we have collectively determined that MHWs are the direct result ...*” – this sentence now opens the sub-section headed **Assessment of the literature on drivers and processes** commencing on Line 199 of the revised manuscript.

line 25-26: wasn't a similar quantitative framework used by Hobday et al. (2018, *Oceanography*) to uniformly and systematically rate the characteristics of MHWs from satellite (NOAA OISST v2) data? What is new here, adding measures for area extent and changing the criteria from 90th percentile to 98th percentile such that the events are more intense, shorter in duration, and cover smaller areas than those previously identified?

Response: Indeed, the recently published paper by Hobday et al. (2018, *Oceanography*) provides a categorisation scheme for MHWs, and analyses some example previous events. The intended contribution from Table 2 in the present submission is the unified analysis of (and thus internal consistency of) intensity, timing and scale of most of the strong MHW events since 1982 supported here by our understanding of the characteristic generating mechanisms. Hobday et al. (2018) only characterises a small subset (10) of these – the well-known events. We also include a wider range of metrics characterising each event.

line 39: here it says that the longest duration events occur in the E. Eq. Pacific associated with El Niño - yet using criteria of Hobday et al. (2018, *Oceanography*) the 2014-2016 MHW in the NE Pacific

falls out as an event that persisted for over 700 days.

Response: We have revised this statement to read “On average, the longest duration MHW events tend to occur in the eastern equatorial Pacific associated with El Niño events, aside from a few longer individual events noted elsewhere in recent years, e.g. the recent 2013-2016 NE Pacific Blob event⁸ and 2015/16 Tasman Sea MHW⁹”. See Lines 120-123 of revised manuscript.

lines 63-64: I find it hard to distinguish between a strong El Niño and a MHW in the ETP. Saying that "the arrival of El Niño pre-conditions the ETP ..." is like saying "an extended spell of extreme hot weather pre-conditions a region for a heat wave", isn't it?

Response: We have modified this sentence now to read “The large increase in SST during El Niño in the eastern tropical Pacific near the Galápagos Islands and the west coast of South America can result in long duration MHW events in this region (Figs. 3 and 4c)²¹”. See Lines 212-214 of revised manuscript.

line 80: with respect to drivers of MHW events in the Indian Ocean - IOD is a dominant mode of SST and wind stress variability in the Indian Ocean, but itself is not a driver. The drivers are the changes in surface wind stress and related air-sea heat fluxes, Ekman transports, and thermocline displacements, and the feedbacks between the IOD pattern of SST and surface wind stress, aren't they?

Response: We refer to our definition of ‘drivers’ which comprise the local processes (e.g. air-sea heat fluxes, advection, etc.), large-scale climate modes (e.g. ENSO, IOD) and teleconnections – see **Methods**. These are collectively the causative mechanisms that combine to produce a MHW event. Hence, the IOD is a contributing causative mechanism for MHWs in this region, where the factors outlined by the reviewer include the local processes which ultimately express the MHW as the extreme anomalous warming at the sea surface. We have simplified our sentence to now read “The IOD, and ENSO via the IOB, are key climate modes contributing to the SST variability in the tropical Indian Ocean^{22,23}” – see Lines 215-216 of the revised manuscript.

lines 82-84: again, saying that the IOD caused the extended duration of the MHW doesn't make sense. The IOD is a mode of coupled ocean-atmosphere climate variation that involves various heat exchange mechanisms and feedbacks between the ocean and atmosphere that promote persistence of the event and its spatial pattern, so it makes more sense to me to describe the events in this way.

Response: We have reworded this sentence to read “The atmospheric processes associated with a negative IOD may have extended the duration of the intense MHW across northern Australia during the 2015/16 El Niño²⁴”. See Lines 220-222 of revised manuscript.

line 100: I would replace “cause” with “favor the seasonal persistence of” ... ENSO related teleconnections are typically defined using monthly or seasonal averages, while “weather” is typically defined with much higher-frequency data (hourly-daily)

Response: Done. See Line 266 of revised manuscript.

line 104: “decreased input of cold Arctic water ...” this is not correct - Arctic water does not influence SSTs in the Gulf of Alaska, and was not a factor in the NE Pacific warm blob in 2013-2016. Anomalously weak cooling of the warm blob region in fall 2013/winter 2014 was due to a combination of anomalously weak vertical mixing, ekman transports, and surface heat fluxes from the ocean to the atmosphere, all related to the persistence of anomalously high atmospheric

pressure that caused anomalously weak winds, clear skies and reduced storminess. The persistence and evolution of the local/regional atmospheric forcing has been the subject of several journal articles, and the nature of the local atmospheric forcing varied over the course of the 2013-2016 period (see your reference 22).

Response: This section has been substantially rewritten to provide a clearer assessment for each region. The subject sentence does not appear in the revised manuscript.

lines 128-134: this is the kind of description that makes a lot of physical sense to me, wherein the local/regional ocean-atmosphere heat exchange processes are described, and the larger scale context is provided.

Response: Acknowledged as a comment.

line 147: insert "local/regional" between "causing SST"

Response: Done. See Line 299 of revised manuscript.

lines 153-154: WBC's are also regions of exceptionally sharp horizontal SST gradients, such that small spatial displacements of the edge of the WBC translate into very large SST anomalies.

Response: We agree that this is an important point and have now also added this to the text (see Lines 305-307 of revised manuscript). Thank you.

lines 165-166: do anomalous poleward currents in EBCs suppress upwelling? Is this specific to remotely forced alongshore currents? Or are there regional wind anomalies contributing to anomalous poleward currents, weaker upwelling and reduced heat flux from the ocean to the atmosphere all at the same time?

Response: An anomalous poleward current in EBCs would be expected to advect warmer water polewards and suppress upwelling, depending on the cross-shelf scale and depth affected by a layer of warmer water. For the case of Ningaloo Niño off Western Australia, both remote and local wind anomalies have been found to contribute to anomalous poleward currents, suppressed upwelling, and coastal warming (e.g. Feng et al., 2013; Kataoka et al., 2014). For the 2011 MHW event, we find that the warming was caused in part (~2/3) due to intensification of the Leeuwin Current and poleward advection and to a lesser extent (~1/3) due to anomalous air-sea heat flux (Benthuisen et al., 2014). On a smaller spatial and temporal scale, anomalous poleward currents have been found to occur off the California coast related to wind relaxations (Washburn et al., 2011) and warming occurred through the arrival of thermal fronts.

References

- Benthuisen J, M Feng and L Zhong. Spatial patterns of warming off Western Australia during the 2011 Ningaloo Niño: Quantifying impacts of remote and local forcing. *Cont. Shelf Res.* **91**, 232-246 (2014).
- Feng M, MJ McPhaden, S-P Xie and J Hafner. La Niña forces Leeuwin Current warming in 2011. *Sci. Rep.* **3**, 1277 (2013).
- Kataoka T, T Tozuka, S Behera and T Yamagata. On the Ningaloo Niño/Niña. *Clim. Dyn.* **43**, 1463-1482 (2014).
- Washburn L, MR Fewings, C Melton and C Gotschalk. The propagating response of coastal circulation due to wind relaxations along the central California coast. *J. Geophys. Res.* **116**(C12028) (2011).

line 203: delete “drivers” from the end of this heading

Response: Done.

line 210: replace “can suppress or enhance” with “are associated with enhanced or suppressed likelihood”

Response: Done (Line 137 of revised manuscript).

line 214: replace “effect” with “association with”

Response: Done (Line 141 of revised manuscript).

line 216: replace “influence on” with “association with”

Response: The sentence has been reworded and the phrase has been changed to “significant relationship with” (Line 143 of revised manuscript).

lines 243-245: consider adding a new set of figures showing the regions having MHW statistics associated with each of the climate modes considered in this study to provide readers a cleaner/clearer view of their associations and potential influences, or don’t show Fig. 4 at all.

Response: We now supplement our new Fig. 3 in the revised manuscript with additional analyses of the relationship between individual climate modes and both the composite analysis of the enhancement and suppression of MHWs together with regressions on SSTA, now presented as a new set of figures in the SOM. Our new Fig. 3 highlights our key points. We hope this now provides a clearer picture of the importance of these modes.

line 251: end this sentence by adding “in the NE Pacific”

Response: Done (Line 182 of revised manuscript).

line 255-256: Would be best to note that statistical relationships do not necessarily indicate causal links much earlier in this article

Response: Done (now first noted on Lines 66-67 of revised manuscript).

lines 262-263: the lack of correlations with climate modes may simply indicate that persistent changes in local or remote forcing (e.g. wind stress) that is not associated with a large scale mode of variability is the key driver

Response: We agree with this reviewer’s point. We have revised this sentence accordingly (see Lines 196-198 of revised manuscript).

lines 276-278: ENSO physics is better described as involving coupled ocean/atmosphere interactions that include changes in winds that impact upwelling intensity, the depth of the thermocline, horizontal advection, o/a heat fluxes and SSTs, and SSTs that impact atmospheric sea level pressure, surface winds, and convection.

Response: We have now reworded this sentence (Lines 232-234 of revised manuscript).

lines 288-291: Di Lorenzo and Mantua (2016) described the NE Pacific MHW as a continuous event that involved various tropical-extratropical teleconnections in the spatial evolution and persistence of the SST anomaly field, but not as an event that faded then re-emerged.

Response: This sentence has been reworded accordingly (Lines 409-411 of revised manuscript).

line 304: with regard to “modes as drivers of MHWs ...” - some modes may be drivers, but others are defined by SST patterns so are diagnostics rather than drivers

Response: We have now reworded “as drivers of” to “in relation to” (Line 424 of revised manuscript).

line 331: what was the cutoff date for your web of science/google scholar search?

Response: The cut-off for our literature search undertaken in this study is February 2016. As such, our consolidated search of the historical literature is contained to the period from January 1950 to February 2016 – while we acknowledge that the literature on MHWs is growing rapidly. We have cited several relevant studies since the cut-off that provide important specific information regarding, for example, global historical trends in MHWs (e.g. Oliver et al., 2018), global projections of MHWs (Frohlicher et al., 2018) and the only significant reported MHW event in a Western Boundary Current and Extension region – the unprecedented 2015/16 Tasman Sea MHW in the East Australian Current Extension (Oliver et al., 2017).

References

- Frohlicher TL, EM Fischer and N Gruber. Marine heatwaves under global warming. *Nature* **560**, 360-364 (2018).
- Oliver ECJ, JA Benthuyzen, NL Bindoff, AJ Hobday, NJ Holbrook, CN Mundy and SE Perkins-Kirkpatrick. The unprecedented 2015/16 Tasman Sea marine heatwave. *Nature Communications*, **8**, 16101 (2017), DOI:10.1038/ncomms16101.
- Oliver ECJ, MG Donat, MT Burrows, PJ Moore, DA Smale, LV Alexander, JA Benthuyzen, M Feng, A Sen Gupta, AJ Hobday, NJ Holbrook, SE Perkins-Kirkpatrick, HA Scannell, SC Straub and T Wernberg. Ocean warming brings longer and more frequent marine heatwaves. *Nature Communications*, **9**, Article number 1324 (2018), DOI:10.1038/s41467-018-03732-9.

Figure 4: too much information - I find these maps to be extremely difficult to interpret, perhaps you could make individual maps for each climate model considered in Supplemental Material

Response: In an effort to provide greater clarity around the figure, we now supplement our new Fig. 3 with separate analyses and Figures provided in the SOM – showing the relationships between individual climate modes from a composite analysis of both enhanced and suppressed MHW occurrences, and regressed on SSTA. We note asymmetries and differences between the analyses of climate mode relationships with MHW occurrences compared with regressions on SSTA that add value to our understanding of the contribution that large-scale climate modes make to the connections with MHW likelihoods. Our additional analyses (figure panels) now provide point-wise estimates of the percentage increase (or decrease) in MHW occurrences in relation to individual climate modes.

Table 1: too much information - I find the information presented here to be extremely difficult to interpret

Response: We have now substantially revised the manuscript text, including providing individual case study region assessment statements (according to climate zone) in the **Results** section outlining the characteristic MHW drivers, so that it more naturally expands upon and clarifies the confidence assessment summary Table 1. We have also redrafted Fig. 2 so that the terminology is now internally consistent with Table 1, and hence the messages more meaningful and informative.

Table 2: I don't think it is meaningful to have 4 distinct MHW events in the Humboldt Current associated with both the 1982/83 and 1997/98 El Niño events, and 3 distinct California MHW events associated with the 1997/98 El Niño. The dynamics of ENSO are such that the basin-scale/global-scale tropical event has a duration of ~6 to 18 months, with peak amplitude and maximum teleconnection to the NE Pacific in the boreal winter months. Each of those sub-seasonal events is really part of the same ENSO event, and if you look at SST anomaly time series for the MHW regions of interest you will not see an end to the warm anomalies, just temporary dips below the 98th percentile

Response: We acknowledge the reviewer's point here. However, to preserve consistency across all analyses, applying an objective way of defining events is important (here, we follow the objective approach of Hobday et al. (2016), albeit scaled for strong events (>98th percentile)). In this case, if no part of the region in question had a MHW that exceeded the threshold for more than 2 days then the MHW was not considered as continuous. As such, we have now added an additional sentence to the caption for Table 2 that reads "Further, some of the closely separated MHW events defined here (e.g. Humboldt/Peru Current region in 1982, 1983 and 1997/98; and the northeast Pacific region in 2013/14 and 2014/15) would be considered a continuous MHW if a weaker threshold was used".

Reference

Hobday AJ, LV Alexander, SE Perkins, DA Smale, SC Straub, ECJ Oliver, J Benthuisen, MT Burrows, MG Donat, M Feng, NJ Holbrook, PJ Moore, HA Scannell, A Sen Gupta and T Wernberg, 2016: A hierarchical approach to defining marine heatwaves. *Progress in Oceanography*, **141**, 227-238, doi:10.1016/j.pocean.2015.12.014.

Table 2: you could add the 2014-2016 record warming of the CCS for "California" (See Jacox et al. 2018, *Bull. of the Am. Met. Soc.*)

Response: We thank the reviewer for bringing this paper to our attention, and readily acknowledge and accept that this could be included. Following our structured literature search cut-off of February 2016, it is indeed reasonable that we could potentially include one or two other case studies since then. However, given the substantiveness of the task for the additional complete and comprehensive reviews of all literature post February 2016, we feel that it is beyond the scope of this study at this point in time to additionally undertake a separate analysis of all literature published in the past couple of years, together with additional quantitative analysis for Table 2. This would take considerable time, with the elapsed additional time amounting to more new literature being published in the intervening period, an endless endeavour.

Table 2: for the Northeast Pacific Ocean, 2013-2016 had two separate events, and not one? This seems at odds with the evolution and persistence of extreme SSTA in that period and how it has been reported in other publications (DiLorenzo and Mantua 2016; Jacox et al. 2018)

Response: Here we have defined periods of particularly strong MHWs (>98th percentile). In this context the Blob appears as two distinct events. In the literature there is a potential separation of these events by their drivers. The initial part of the event was associated with a persistent high-

pressure system (and was located further offshore), while the latter part of the event was associated with a tropical teleconnection (and located along the coast).

Nate Mantua
NOAA/NMFS/SWFSC
Santa Cruz, CA

Reviewer #2

Thank you for the thoughtful and constructive review of our manuscript. We provide our point-by-point responses (in black) to the comments and issues raised by the reviewer (in blue) and have revised our manuscript accordingly.

Earlier studies, some of which appeared this year, have begun to document the rise in amplitude, occurrence, and duration of marine heatwaves (MHW) around the world's oceans. However, while these statistics of sea surface temperature extremes (SSTe) provide useful spatial and temporal information on a global-scale, they lack information about the drivers and mechanisms underlying the emergence of MHW. This manuscript is an important first attempt to assemble an inventory of documented MHW in the literature and examine what are the dominant/recurring mechanisms and drivers of extreme warm events.

Response: We appreciate the reviewer's recognition of the value of our study.

Given the regional dependence of the majority of these events, it is hard for the manuscript to provide real novel insights on the local mechanisms that trigger SSTe other than invoking all the usual oceanographic drivers (e.g. heat fluxes, advection, upwelling, winds). However, the authors do attempt to establish on a global-scale a relation between the SSTe and dominant climate modes such as ENSO, PDO, AMO, and others. They do so by both "text-mining" the database of papers documenting the individual events and by performing a statistical analysis of the relation between observed SST changes and the climate modes. Although the analysis does not lead to very clear and coherent attributions (e.g. Figure 4 ultimately is pretty noisy), it does establish the supporting evidence that the occurrence of MHW is significantly linked to the phase of several dominant climate modes. This is an important statement, which has been made in a lot of the individual studies but never confirmed as a global characteristic of most MHW. Such a finding provides the basis for future studies that attempt to exploit more systematically the dynamics of large-scale climate variability as a proxy to understand and predict the statistics of MHW, and may lead to interesting frameworks that relate large-scale climate variance to regional extremes. I also find that this "atlas" (e.g. tables 1 and 2) of MHW and their drivers (e.g. climate modes) appears to be a valuable first step towards synthesizing in one place what is known about drivers with the appropriate references.

Response: We have now undertaken further analysis to better extract the contribution from individual climate modes to MHW occurrences, that is, complementary to relationships between climate modes and SSTA more broadly. We believe this additional analysis makes clearer the large-scale mechanisms that implicitly modulate MHWs locally and are underpinned by our new Fig. 3. Specifically, these additional individual mode analyses – with all provided in the SOM – demonstrate that MHWs may be enhanced or suppressed in regions where SSTA correspondingly may not be. That is, there is an asymmetry in this response that is not necessarily expected or intuitive. We believe this analysis adds value to our understanding of the large-scale drivers of MHW occurrences expressed at the grid-scale. Our new Fig. 4 should hopefully provide greater clarity also to the

analysis, highlighting the importance of individual modes to MHW enhancement or suppression across all of the 22 case study regions.

We have also now restructured our manuscript based on the organising principle that MHW occurrences (identified here as SSTA tendencies expressed through persistent temperature extremes) are underpinned by local mixed layer temperature budgets – see Equation (1) plus the local scale processes in Fig. 2 corresponding to those presented in Table 1, based on our confidence assessment of the relevant MHW forcing mechanisms reported in the literature. We hope that with this manuscript restructure and the new figures, the novelty of our study is more apparent, the key messages clearer, and the new knowledge better communicated.

Overall my inclination is to recommend publication with some revisions detailed below. However, I also have to note some mixed feeling. I anticipate that other reviewers will have different thoughts on the scientific value of this manuscript. In all fairness, this manuscript does not constitute a significant advance in our understanding given that a relationship between the phase of the climate modes and MHW is to be expected and already documented in many of the studies used from the article database. As such, one could argue whether this contribution would warrant a Nature publication. On the other hand, I would recommend publication because such a synthesis of the existing knowledge in the area of ocean extremes/drivers does add real value to the ongoing debate (e.g. it will likely be cited a lot and become a useful reference) and constitutes an enabler (e.g. for future studies) for advancing our understanding of the relation between ocean extreme and large-scale climate dynamics on global-scale. Now, it is true that the climate modes alone do not represent a “real” mechanistic framework because these modes are statistical proxies of the large-scale climate dynamics. I am also sure that after completing this study, one could go back to the drawing board and re-design certain approaches. Nevertheless, I commend the authors for this first step towards exploring mechanistic drivers of MHW on a global-scale. I think the manuscript brings sufficient rigor in the approaches and ultimately delivers a valuable “global” summary of our understanding even if using a simplistic/fragmented view of the climate mechanisms.

Response: We are delighted the reviewer is positive overall about our manuscript, while we acknowledge their points raised. Regarding the MHW relationship with climate mode phase, we believe our additional analysis that contribute to our new Fig. 3 (as well as the new Fig. 4) – i.e. deconstructing the relationships between individual climate mode and MHW occurrences, and which is provided in the revised SOM – should now provide a more complete and compelling case for the importance and relevance of climate modes in modulating MHW occurrences locally, recognising that the relationship between the same climate modes and SSTA may be quite different.

We believe that the significant advance in our understanding of MHW drivers contributed by our manuscript is collectively due to (1) the global scale of this study, (2) the systematic characterisation of large-scale drivers, teleconnections and local processes that cause MHWs through our confidence assessment of the historical peer-reviewed literature classified within geographic, climatic and oceanographic zones – noting that our study represents a critical assessment and not simply a review, and (3) the grid-scale analysis of MHW enhancement or suppression significantly related to large-scale climate mode phase, providing implicit information about the potential predictability of MHWs and direction for future prediction studies.

Reviewer #3

Thank you for the thoughtful and constructive review of our manuscript. We provide our point-by-point responses (in black) to the comments and issues raised by the reviewer (in blue) and have revised our manuscript accordingly.

This is a literature review and synthesis of Marine Heat Waves globally. The manuscript summarizes the characteristics of heat waves appearing around the world since 1982, and a global SST data set is used to then relate heat waves to known modes of ocean-atmosphere variability. There is a table that summarizes findings from the literature geographically, and also a figure that relates phases of different modes with temperature increases or decreases. In addition, a figure which schematically places heat waves into the context of time and space scales of ocean variability is included.

The scope of this work and the synthesis is sound, but is somewhat limited because only sea surface temperature is analyzed. Since this is primarily a review of previous findings, with further analysis of SST data, there is not a comprehensive heat budget presented, or consideration of the vertical scale of the temperature anomalies. There is a statement (lines 375-376) that sub-surface data are sparse, and that ecological impacts are generally realized in the upper ocean. While the SST analysis is enlightening, a more thorough treatment would include a full heat budget analysis of the surface mixed layer. Since the subject is of global extent, this is impossible with the lack of sub-surface observations. However, given the wide range of space and time scales in Figure 2, there needs to be some discussion of how the SST analysis relates to changes in the mixed layer or upper ocean. Some of the processes that are included in Figure 2, such as ENSO or Kelvin waves, do have known signals in terms of changes in the depth of the mixed layer or changes in stratification. Ecosystem effects are concentrated in the upper ocean, but mixed layer properties are likely more indicative than SST. Some discussion is needed of how the heat waves might be affecting mixed layers. This can be a general statement, but the consideration of only SST is a big limitation and needs a more general context for upper ocean changes.

Response: We have now endeavoured to provide a much clearer narrative by restructuring our manuscript so that the mixed layer temperature (MLT) budget is the organising principle for understanding marine heatwave (MHW) occurrences locally. Fundamental to this restructuring, we have now identified the MLT equation (Equation 1) up-front in the introductory portion of our revised manuscript as the framework for our understanding of the dynamics/thermodynamics that underpin MHWs caused directly by the local fluxes, and that connects to the remote large-scale climate modes and their teleconnections which act to modulate the MHW response. A schematic of this is provided by Fig. S1, which we retain as per the originally submitted manuscript.

In our revised manuscript, we now ensure that the relevant MHW processes in Fig. 2 (space-time scale schematic) correspond more directly and informatively with the driver/process characteristics reported in Table 1 (categorised by time scale and region) – resulting from our confidence assessment analysis of the mechanisms causing MHWs reported in the peer-reviewed literature. Hence, while our study doesn't separately explicitly conduct a suite of temperature budget analyses (which would depend on subsurface data coverage that is historically unavailable, and if it were realistically possible would represent an enormous undertaking), our confidence assessment does implicitly take account of the mechanistic analyses in the MLT and their drivers undertaken within the individual studies reported in the peer-reviewed literature.

We have now added some words in the introduction to reinforce that our statistical analysis of the satellite data is based on MHWs identified in sea surface temperature (SST) data. We point out our recognition of the importance of the ML in relation to diagnosing processes, as well as MHW impacts on ecology, and that we have synthesised previously reported explicit analyses of MHW causative MLT processes in our overarching confidence assessment of the historical peer-reviewed literature.

We added the following to our introduction:

“A useful lens for understanding the formation, maintenance and decay of MHWs is the upper ocean mixed layer heat budget, which includes the local processes responsible for changes in surface ocean temperatures. For MHW events, the dominant contributions to the temperature tendency within the mixed layer^{13–15} are given by

$$\frac{\partial T}{\partial t} = -\frac{1}{H} \int_{-H}^0 (\mathbf{u} \cdot \nabla_h T) dz + \frac{Q}{\rho C_p H} + \text{residual} \quad (1)$$

where the first term on the right hand side (RHS) is the horizontal advection (ocean advective fluxes) owing to the depth-dependent horizontal velocity vector \mathbf{u} and temperature T , the second term is the contribution for the net air-sea heat flux Q , where ρ is the average seawater density, C_p is the specific heat capacity of seawater ($4000 \text{ J kg}^{-3} \text{ }^\circ\text{C}^{-1}$), H is the mixed layer depth, and the residual third term includes horizontal eddy heat fluxes and the heat flux at the bottom of the mixed layer owing to vertical diffusion, entrainment, and vertical advection. To generate MHWs, the physical processes on the RHS of (1) have to contribute a sufficiently large net positive temperature tendency to increase the surface ocean temperature over a threshold⁶. Oceanic advective fluxes include unusually intense poleward flow, associated with the transport of warmer waters, enhanced heat advection by boundary currents or reduced turbulent mixing. MHW favourable air-sea heat fluxes include contributions from anomalous net downward surface radiation, with little cloud cover and convective air-sea heat fluxes, and with suppressed latent heat loss from anomalously weak surface winds. While Equation (1) explicitly describes temperature variations that extend through the mixed layer, MHWs may extend deeper. The intensity of surface intensified MHWs may also depend on the state of the subsurface ocean, such as shallowing of the mixed layer depth^{10,15}.

In the vast majority of research conducted to date, and in our global analysis presented here, MHWs have been identified and characterised based on sea surface temperature (SST). This focus reflects the relative scarcity of data from the subsurface ocean, and the reported ecological impacts that are largely prevalent in the upper ocean where biological productivity is greatest. Key mechanisms that drive temperature changes in the mixed layer may include local internal variability (e.g. eddy instabilities) or large-scale modes of climate variability that act to modulate the local conditions – either from (1) local sources, for example, extreme ocean-atmosphere coupled feedbacks in the eastern equatorial Pacific during extreme El Niño – Southern Oscillation (ENSO) events, or (2) remote sources via ‘teleconnection’ mechanisms by, for example, the propagation of planetary waves in the atmosphere (that give rise to distant changes in the surface winds, cloud cover, etc.) or ocean (that can change the depth of the thermocline and drive remote circulation changes).”

There needs to be a statement in the main text that the SST analysis includes only events within the 98th percentile or above. The common usage before has been 90th percentile. I did not see this until the very end of the supplemental material. This needs to be featured in the text.

Response: In our responses, please note that the identified line numbers refer to our clean version of the revised manuscript, rather than the track-change version (where the track-changes are extensive).

We have added the following sentence early in our revised manuscript (Lines 91-94) *“These characteristics are estimated based on a consistent MHW definition⁶ which allows for comparability (note a simple rescaling was uniformly applied (see Methods) to identify strong MHW events) given that the MHWs identified in our literature search have been reported using a diverse set of criteria”.*

Figure 4 is practically unintelligible. The intent of the figure and the analysis that went into it is admirable, but between the large number of modes/colors, the extreme pixel to pixel variations, and the small size of the figure, it was very difficult to interpret. Just looking at the north Atlantic for example, it is hard to see any signal in the central part of the basin away from the NAO dominant regions in the north and south. Don't know if this needs to be split into more panels (eastern and western hemisphere) but it is just too much information.

Response: We acknowledge that this is a complex figure that summarises a large amount of information. Nevertheless, while indeed noisy in places, it also indicates large coherent areas where certain modes clearly dominate, providing useful information. Regions with lots of noise simply highlight areas where no single climate mode dominates. As suggested, to supplement this figure we now also present the analysis by individual climate mode that underpins this summary figure. Some of these are included in our new Fig. 3, and others are also provided in the SOM, showing the proportion of time experiencing MHW conditions during positive and negative phases of each of the modes (where the increase or decrease exceeds what would be expected from random sampling).

Essentially, to more clearly demonstrate the importance of climate modes to MHWs, and the relevant contribution, we deconstructed our composite analysis further. This approach also responds to a concern raised by Reviewer #1. Specifically, in the SOM we now show the relationships between individual modes of climate variability and, separately, (a) SSTA, and (b) MHW occurrences. For the MHW analyses, the percentages are relative to the median number of MHW days (shown in new Fig. 3), e.g. in a region where we expect MHWs to occur 10% of the time, an 80% increase means we would expect MHWs to occur 18% of the time associated with that phase of the associated mode.

We believe our results from the deconstructed analysis should be more illuminating of the contribution from the new Fig. 3. In the SOM, our analysis also demonstrates that SSTA and MHW occurrences are not characterised in the same way in relation to these climate modes. We find clear asymmetries (and regional differences) in the spatial distributions of percentage increase or decrease in MHW occurrences across the globe that are very different from those distributions found from climate mode regressions on +/- SSTA alone. Examples of the contributions from this analysis are seen in the cases of the Southern Annular Mode and ENSO comparisons between SSTA and MHW occurrences.

These figures effectively characterise the potential for predictability of MHWs associated with climate modes. That is, given a perfect forecast, these figures provide spatial maps of the distribution of percentage likelihoods of MHW increased or decreased occurrence in relation to each phase of the individual climate modes.

On the whole, the analysis is sound, the geographical coverage is extensive, and the attempt to establish dominant drivers is novel and important. Providing a better context for the results presented by having some discussion of vertical scales/mixed layer variability would greatly improve the impact of the analysis and synthesis. Efforts such as the ARGO floats have provided a lot of global scale information on temperature fields, and so future efforts to study heat waves on large space and time scales will need to include subsurface information in addition to SST.

Response: We are really pleased the reviewer notes the novelty and importance of our study. Indeed, datasets are now becoming available that will allow us to understand the three-dimensional structure of MHWs. Ocean reanalysis products provide spatially and temporally continuous views of the subsurface ocean and ARGO profiles provide us with unprecedented subsurface coverage,

although these have only been available since early this century. We have now added these points to our discussion. See Lines 444-448 of our revised manuscript.

The results are significant, and with the above issues addressed this would be worthy of publication.

We are delighted that the reviewer finds our manuscript to be potentially worthy of publication. We hope that our responses above satisfy the reviewer's concerns.

Reviewer #1 (Remarks to the Author):

R2 MHW Nature

Overall I find this revised manuscript to be much improved over the previous version. I appreciate the extensive revisions the authors have carried out, and especially like the new Fig. 4 and the supplemental figures S3-S15. I think that this is very well written and informative, and will make a nice contribution to the rapidly growing literature on marine heatwaves. I recommend that this manuscript be accepted with a request for minor revisions.

Below I have a few specific comments, questions, and recommendations for revisions that should be easy for the authors to respond to.

Main text and figures:

line 8: I would revise “teleconnections that drive MHWs regionally” to “teleconnections that are associated with MHWs regionally”

line 132: Is your literature review for 1982-2016, or 1982-2017 (make sure text here is consistent with Methods section and the caption for Table 2)

line 241: insert “eastward propagating equatorial” before “Kelvin”

lines 364 vs 372: Here, your summary suggests that MHWs in eastern boundary current regions are typically associated with modes of tropical climate variations. However, the whole point of ref 70 is that there is a local/regional mode of coupled air-sea interactions off Baja and California in summer that is independent of the much broader scale ENSO phenomenon. A more recent article by Fiedler and Mantua (2017) also emphasizes the importance of local/regional atmospheric forcing for the broader CCLME warm and cold events that is only modestly correlated with ENSO. (See 1.

Fiedler, P., and N.J. Mantua, 2017. How are warm and cool years in the California Current related to ENSO? JGR Oceans. Doi: 10.1002/2017JC013094)

line 405: add “, both locally and remotely because of eastward propagating equatorial Kelvin waves,
” after “westward flow of water”

Lines 409-411: repeat or expand upon this sentence in the SI section discussing the 2013-16 northeast Pacific MHW

lines 433-434: I would say that your results reveal the potential for compounding influences, but not that such compounding influences exist. What you've shown are associations between regional MHW statistics and multiple climate modes in some regions. Yet, these climate modes are in many/most cases not independent of each other, and it isn't clear if there are compounding influences. Investigating the potential for compounding influences could make for a good follow up study to this one.

lines 505-508: It might be worthwhile to note that each of the climate modes considered here are not necessarily independent from each other, and in fact there are clear statistical and dynamical connections between many of them.

Fig 1 caption: what you have labelled "California Current" is really the southern extreme of the California Current, and might be better labelled as a "Baja California" region.

Table 2: I'm surprised that your California MHW case studies don't include anything for MHW days during the 2014-2016 event - is that too recent to be in your literature review? It is, to date, the grand-daddy NE Pacific/California Current warm event of them all! Likewise, your NE Pacific region has 2013/14, but nothing for 2015 (or 2016). Again, are those periods simply too recent for your literature review?

Table 2 typos: last column for Galapagos Is. and Bay of Bengal has "/" where they do not belong

supplemental information

line 118-119: It might be worth noting that the "California Niño/Niña" is centered off Baja California, very much in keeping with the region indicated in your Figure 1. Your Table S4 lists California Current region MHW case studies includes events identified for the broader CCS in Fiedler and Mantua 2017, but lacks the events they identified in 1967-68, 1977-78, 1979-80, 2014-15, and 2015-16 (see their Fig. 5). Again, I recognize that this publication may have come too late to be included in your literature review, and their analysis was not framed around MHW criteria.

line 368: after "remote wind changes" add "that generate eastward propagating downwelling equatorial Kelvin waves"

line 401: add "atmospheric" before "teleconnections"

lines 529-550: This section has lots of repetition and can be streamlined for a better summary of the local/regional mechanisms that contributed to the NW Atlantic MHW event in 2011/12, and how those mechanisms were influenced by larger scale atmospheric blocking associated with jet stream anomalies

line 555: This summary for the NE Pacific is focused on fall 2013 through winter 2014, and does not summarize the evolution of the offshore "warm blob" into the NEP Arc pattern in fall 2014/winter 2015. See DiLorenzo and Mantua (2016) for a description of the spatio-temporal evolution of this event, and how the regional atmospheric forcing over the NE Pacific was related to large-scale tropical and extratropical modes of Pacific climate variability included in your study.

Nate Mantua

NOAA/NMFS Santa Cruz, CA

Reviewer #3 (Remarks to the Author):

This manuscript has been substantially re-written and improved. The use of the heat balance for the mixed layer as the initial organizing principle makes the organization of the synthesis fall into logical categories. In particular, the division of the statistical analysis into MHWs versus previously analyzed SST patterns clarifies some important relationships with climate modes.

The breakdown into regional categories for the discussion with the emphasis on the modes of variability was sensible in the new framework. I was particularly gratified to see that the first panel of Figure 3 now makes the complicated patterns in the second panel of Figure 3 (formerly Figure 4 of the original manuscript, I think) much more comprehensible.

The tables, while complicated, convey important information and will be used to further understanding in all of the region sub-categories.

My only minor comment is that it appears that the resolution of Figure 3 seems to be a bit grainy, higher resolution in the final version would be helpful.

I think the new version addresses all of the issues from the original review. The new organizing principle works well, and while the patterns and statistical relationships are still complex, the manuscript reads well.

Excellent job in improving the manuscript, and I find this acceptable for publication with the one minor comment on the figure in mind.

RE: Manuscript NCOMMS-18-17432B

RESPONSES TO THE EDITOR AND REVIEWERS

29 March 2019

We sincerely wish to again thank the Editor and the three reviewers for their thoughtful and constructive reviews of our manuscript entitled “A global assessment of marine heatwaves and their drivers”. We are delighted that this manuscript has been accepted, in principle, providing that we satisfactorily address the remaining concerns raised by the reviewers, and the requirements outlined by the Editor. Please find attached our second revision (R2) manuscript, which has been carefully revised accordingly.

We note that through the review process, our manuscript text grew with the intention to provide a much stronger and clearer structure with greater clarity around our results and findings. As a result, our revised manuscript grew to 5,926 words (Introduction, Results and Discussion) and 87 references, which exceeds the recommended limit. We hope this can be facilitated. We provide a copy of our completed compliance checklist as requested.

We have addressed all other compliance requirements raised within the manuscript and provide responses to these requests (in the comments column) of the track-change version of our R2 manuscript. We also provide production-quality versions of our figures as separate files. We hope that Tables 1 and 2 can be published on single pages in the typesetting. Please advise if this is likely to pose any problems.

We provide here our point-by-point responses (in black) to the remaining issues/comments raised by the two reviewers in the second round of reviews (in blue), and we have revised our R2 manuscript accordingly. In our responses to the reviewers below, please note that the identified line numbers refer to our **clean** version of the revised manuscript.

REVIEWERS' COMMENTS:

Reviewer #1 (Remarks to the Author):

R2 MHW Nature

Overall I find this revised manuscript to be much improved over the previous version. I appreciate the extensive revisions the authors have carried out, and especially like the new Fig. 4 and the supplemental figures S3-S15. I think that this is very well written and informative, and will make a nice contribution to the rapidly growing literature on marine heatwaves. I recommend that this manuscript be accepted with a request for minor revisions.

Response: We would like to again sincerely thank the reviewer for their very thoughtful and constructive reviews of our manuscript, which have led to substantial improvements in the structure, quality and readability of the paper.

Below I have a few specific comments, questions, and recommendations for revisions that should be easy for the authors to respond to.

Main text and figures:

line 8: I would revise “teleconnections that drive MHWs regionally” to “teleconnections that are associated with MHWs regionally”

Response: Done (see line 52 of revised manuscript).

line 132: Is your literature review for 1982-2016, or 1982-2017 (make sure text here is consistent with Methods section and the caption for Table 2)

Response: Our comprehensive and systematic literature assessment includes publications from 1950 to February 2016. Our analysis of the satellite SST data is based on the available data from 1982-2016. We have revised the abstract and main text to make this clear (see lines 49-51, 118 and 502-503 of revised manuscript).

We acknowledge that the literature on marine heatwaves (MHWs) is growing rapidly. As such, we considered it important to cite a few additional relevant studies since the cut-off that provide specific information regarding, for example, global historical trends in MHWs (e.g. Oliver et al. 2018), global projections of MHWs (Froelicher et al. 2018) and the only significant reported MHW event in a Western Boundary Current and Extension region – the unprecedented 2015/16 Tasman Sea MHW in the East Australian Current Extension (Oliver et al. 2017).

line 241: insert “eastward propagating equatorial” before “Kelvin”

Response: Done (see lines 285-286 of revised manuscript).

lines 364 vs 372: Here, your summary suggests that MHWs in eastern boundary current regions are typically associated with modes of tropical climate variations. However, the whole point of ref 70 is that there is a local/regional mode of coupled air-sea interactions off Baja and California in summer that is independent of the much broader scale ENSO phenomenon. A more recent article by Fiedler and Mantua (2017) also emphasizes the importance of local/regional atmospheric forcing for the broader CCLME warm and cold events that is only modestly correlated with ENSO. (See 1.

Fiedler, P., and N.J. Mantua, 2017. How are warm and cool years in the California Current related to ENSO? JGR Oceans. Doi: 10.1002/2017JC013094)

Response: We acknowledge this point. Our summary paragraph for eastern boundary MHWs has now been revised to better reflect the role of, and added complexity from, eastern boundary Niños on MHWs that may be independent of (or only loosely dependent on) ENSO. See lines 416-430 of revised manuscript.

line 405: add “, both locally and remotely because of eastward propagating equatorial

Kelvin waves, “ after “westward flow of water”

Response: Done (see lines 451-452 of revised manuscript).

Lines 409-411: repeat or expand upon this sentence in the SI section discussing the 2013-16 northeast Pacific MHW

Response: We have now included this point in the revised Supplementary Information (see page 13 lines 10-13 of paragraph 3).

lines 433-434: I would say that your results reveal the potential for compounding influences, but not that such compounding influences exist. What you've shown are associations between regional MHW statistics and multiple climate modes in some regions. Yet, these climate modes are in many/most cases not independent of each other, and it isn't clear if there are compounding influences. Investigating the potential for compounding influences could make for a good follow up study to this one.

Response: We agree and have now revised our wording in the text to better reflect this important point (see lines 480-482 of revised manuscript).

lines 505-508: It might be worthwhile to note that each of the climate modes considered here are not necessarily independent from each other, and in fact there are clear statistical and dynamical connections between many of them.

Response: Done (see lines 558-560 of revised manuscript).

Fig 1 caption: what you have labelled “California Current” is really the southern extreme of the California Current, and might be better labelled as a “Baja California” region.

Response: We have now amended the Figure 1 caption accordingly.

Table 2: I'm surprised that your California MHW case studies don't include anything for MHW days during the 2014-2016 event - is that too recent to be in your literature review? It is, to date, the grand-daddy NE Pacific/California Current warm event of them all! Likewise, your NE Pacific region has 2013/14, but nothing for 2015 (or 2016). Again, are those periods simply too recent for your literature review?

Response: We do show the northeast Pacific event that stretched through 2015 largely north of 40°N. However, the reviewer is correct to point out that overlapping with this there was a major event that affected coastal regions largely between 20°N and 40°N. We have re-analysed the sea surface temperature data for the 2015/16 period and included the relevant event characteristics for this period against the Baja California case study region in Table 2.

Table 2 typos: last column for Galapagos Is. and Bay of Bengal has “/” where they do not belong

Response: Typos have been corrected in Table 2 and Supplementary Table 2 of the revised manuscript. In these tables, “/” are now correctly replaced by “.”

supplemental information

line 118-119: It might be worth noting that the “California Niño/Niña” is centered off Baja California, very much in keeping with the region indicated in your Figure 1. Your Table S4 lists California Current region MHW case studies includes events identified for the broader CCS in Fiedler and Mantua 2017, but lacks the events they identified in 1967-68, 1977-78, 1979-80, 2014-15, and 2015-16 (see their Fig. 5). Again, I recognize that this publication may have come too late to be included in your literature review, and their analysis was not framed around MHW criteria.

Response: We have now renamed this region as “California Current and off Baja California”. While we would really like to include more recent papers, we have endeavoured to be as consistent as possible with our analysis. This renaming of the region should now be clearer about the geography and thus better characterise the region where California Niños/Niñas are influential. See bottom of page 11 of revised Supplementary Information (SI).

line 368: after “remote wind changes” add “that generate eastward propagating downwelling equatorial Kelvin waves”

Response: Done (see page 24 lines 3-4 of revised SI).

line 401: add “atmospheric” before “teleconnections”

Response: Done (see page 25 last line of paragraph 2 of revised SI).

lines 529-550: This section has lots of repetition and can be streamlined for a better summary of the local/regional mechanisms that contributed to the NW Atlantic MHW event in 2011/12, and how those mechanisms were influenced by larger scale atmospheric blocking associated with jet stream anomalies

Response: We have streamlined this section and provide a more concise description for the northwest Atlantic MHW in our revised manuscript (see page 31 paragraph 2 of revised SI).

line 555: This summary for the NE Pacific is focused on fall 2013 through winter 2014, and does not summarize the evolution of the offshore “warm blob” into the NEP Arc pattern in fall 2014/winter 2015. See DiLorenzo and Mantua (2016) for a description of the spatio-temporal evolution of this event, and how the regional atmospheric forcing over the NE Pacific was related to large-scale tropical and extratropical modes of Pacific climate variability included in your study.

Response: We now explicitly cite the Di Lorenzo and Mantua (2016) study regarding the multi-year persistence of the 2014/15 event in the revised Supplementary Information in both the Baja California and northeast Pacific sub-sections (see page 13 lines 10-13 of paragraph 3, page 32 last line and page 33 first two lines of revised SI).

Nate Mantua
NOAA/NMFS Santa Cruz, CA

Reviewer #3 (Remarks to the Author):

This manuscript has been substantially re-written and improved. The use of the heat balance for the mixed layer as the initial organizing principle makes the organization of the synthesis fall into logical categories. In particular, the division of the statistical analysis into MHWs versus previously analyzed SST patterns clarifies some important relationships with climate modes.

The breakdown into regional categories for the discussion with the emphasis on the modes of variability was sensible in the new framework. I was particularly gratified to see that the first panel of Figure 3 now makes the complicated patterns in the second panel of Figure 3 (formerly Figure 4 of the original manuscript, I think) much more comprehensible.

The tables, while complicated, convey important information and will be used to further understanding in all of the region sub-categories.

Response: We would like to again sincerely thank the reviewer for their very thoughtful and constructive reviews of our manuscript, which have led to substantial improvements in the structure, quality and readability of the paper.

My only minor comment is that it appears that the resolution of Figure 3 seems to be a bit grainy, higher resolution in the final version would be helpful.

Response: The Figure 3 resolution is consistent with the grid-scale of our analysis. We now provide Figure 3 in .pdf format with our revised manuscript.

I think the new version addresses all of the issues from the original review. The new organizing principle works well, and while the patterns and statistical relationships are still complex, the manuscript reads well.

Excellent job in improving the manuscript, and I find this acceptable for publication with the one minor comment on the figure in mind.

Response: Thank you.

REVIEWERS' COMMENTS:

Reviewer #1 (Remarks to the Author):

R2 MHW Nature

Overall I find this revised manuscript to be much improved over the previous version. I appreciate the extensive revisions the authors have carried out, and especially like the new Fig. 4 and the supplemental figures S3-S15. I think that this is very well written and informative, and will make a nice contribution to the rapidly growing literature on marine heatwaves. I recommend that this manuscript be accepted with a request for minor revisions.

Response: We would like to again sincerely thank the reviewer for their very thoughtful and constructive reviews of our manuscript, which have led to substantial improvements in the structure, quality and readability of the paper.

Below I have a few specific comments, questions, and recommendations for revisions that should be easy for the authors to respond to.

Main text and figures:

line 8: I would revise “teleconnections that drive MHWs regionally” to “teleconnections that are associated with MHWs regionally”

Response: Done (see line 52 of revised manuscript).

line 132: Is your literature review for 1982-2016, or 1982-2017 (make sure text here is consistent with Methods section and the caption for Table 2)

Response: Our comprehensive and systematic literature assessment includes publications from 1950 to February 2016. Our analysis of the satellite SST data is based on the available data from 1982-2016. We have revised the abstract and main text to make this clear (see lines 49-51, 118 and 502-503 of revised manuscript).

We acknowledge that the literature on marine heatwaves (MHWs) is growing rapidly. As such, we considered it important to cite a few additional relevant studies since the cut-off that provide specific information regarding, for example, global historical trends in MHWs (e.g. Oliver et al. 2018), global projections of MHWs (Froelicher et al. 2018) and the only significant reported MHW event in a Western Boundary Current and Extension region – the unprecedented 2015/16 Tasman Sea MHW in the East Australian Current Extension (Oliver et al. 2017).

line 241: insert “eastward propagating equatorial” before “Kelvin”

Response: Done (see lines 285-286 of revised manuscript).

lines 364 vs 372: Here, your summary suggests that MHWs in eastern boundary current regions are typically associated with modes of tropical climate variations. However, the whole point of ref 70 is that there is a local/regional mode of coupled air-sea interactions off Baja and California in summer that is independent of the much broader scale ENSO phenomenon. A more recent article by Fiedler and Mantua (2017) also emphasizes the importance of local/regional atmospheric forcing for the broader CCLME warm and cold events that is only modestly correlated with ENSO. (See 1.

Fiedler, P., and N.J. Mantua, 2017. How are warm and cool years in the California Current related to ENSO? JGR Oceans. Doi: 10.1002/2017JC013094)

Response: We acknowledge this point. Our summary paragraph for eastern boundary MHWs has now been revised to better reflect the role of, and added complexity from, eastern boundary Niños on MHWs that may be independent of (or only loosely dependent on) ENSO. See lines 416-430 of revised manuscript.

line 405: add “, both locally and remotely because of eastward propagating equatorial

Kelvin waves, “ after “westward flow of water”

Response: Done (see lines 451-452 of revised manuscript).

Lines 409-411: repeat or expand upon this sentence in the SI section discussing the 2013-16 northeast Pacific MHW

Response: We have now included this point in the revised Supplementary Information (see lines 136-138).

lines 433-434: I would say that your results reveal the potential for compounding influences, but not that such compounding influences exist. What you've shown are associations between regional MHW statistics and multiple climate modes in some regions. Yet, these climate modes are in many/most cases not independent of each other, and it isn't clear if there are compounding influences. Investigating the potential for compounding influences could make for a good follow up study to this one.

Response: We agree and have now revised our wording in the text to better reflect this important point (see lines 480-482 of revised manuscript).

lines 505-508: It might be worthwhile to note that each of the climate modes considered here are not necessarily independent from each other, and in fact there are clear statistical and dynamical connections between many of them.

Response: Done (see lines 558-560 of revised manuscript).

Fig 1 caption: what you have labelled “California Current” is really the southern extreme of the California Current, and might be better labelled as a “Baja California” region.

Response: We have now amended the Figure 1 caption accordingly.

Table 2: I'm surprised that your California MHW case studies don't include anything for MHW days during the 2014-2016 event - is that too recent to be in your literature review? It is, to date, the grand-daddy NE Pacific/California Current warm event of them all! Likewise, your NE Pacific region has 2013/14, but nothing for 2015 (or 2016). Again, are those periods simply too recent for your literature review?

Response: We do show the northeast Pacific event that stretched through 2015 largely north of 40°N. However, the reviewer is correct to point out that overlapping with this there was a major event that affected coastal regions largely between 20°N and 40°N. We have re-analysed the sea surface temperature data for the 2015/16 period and included the relevant event characteristics for this period against the Baja California case study region in Table 2.

Table 2 typos: last column for Galapagos Is. and Bay of Bengal has “/” where they do not belong

Response: Typos have been corrected in Table 2 and Supplementary Table 2 of the revised manuscript. In these tables, “/” are now correctly replaced by “.”

supplemental information

line 118-119: It might be worth noting that the “California Niño/Niña” is centered off Baja California, very much in keeping with the region indicated in your Figure 1. Your Table S4 lists California Current region MHW case studies includes events identified for the broader CCS in Fiedler and Mantua 2017, but lacks the events they identified in 1967-68, 1977-78, 1979-80, 2014-15, and 2015-16 (see their Fig. 5). Again, I recognize that this publication may have come too late to be included in your literature review, and their analysis was not framed around MHW criteria.

Response: We have now renamed this region as “California Current and off Baja California”. While we would really like to include more recent papers, we have endeavoured to be as consistent as possible with our analysis. This renaming of the region should now be clearer about the geography and thus better characterise the region where California Niños/Niñas are influential. See line 94 of revised Supplementary Information (SI).

line 368: after “remote wind changes” add “that generate eastward propagating downwelling equatorial Kelvin waves”

Response: Done (see lines 314-315 of revised SI).

line 401: add “atmospheric” before “teleconnections”

Response: Done (see line 339 of revised SI).

lines 529-550: This section has lots of repetition and can be streamlined for a better summary of the local/regional mechanisms that contributed to the NW Atlantic MHW event in 2011/12, and how those mechanisms were influenced by larger scale atmospheric blocking associated with jet stream anomalies

Response: We have streamlined this section and provide a more concise description for the northwest Atlantic MHW in our revised manuscript (see lines 440-450 of revised SI).

line 555: This summary for the NE Pacific is focused on fall 2013 through winter 2014, and does not summarize the evolution of the offshore “warm blob” into the NEP Arc pattern in fall 2014/winter 2015. See DiLorenzo and Mantua (2016) for a description of the spatio-temporal evolution of this event, and how the regional atmospheric forcing over the NE Pacific was related to large-scale tropical and extratropical modes of Pacific climate variability included in your study.

Response: We now explicitly cite the Di Lorenzo and Mantua (2016) study regarding the multi-year persistence of the 2014/15 event in the revised Supplementary Information in both the Baja California and northeast Pacific sub-sections (see lines 136-138 and 471-473 of revised SI).

Nate Mantua
NOAA/NMFS Santa Cruz, CA

Reviewer #3 (Remarks to the Author):

This manuscript has been substantially re-written and improved. The use of the heat balance for the mixed layer as the initial organizing principle makes the organization of the synthesis fall into logical categories. In particular, the division of the statistical analysis into MHWs versus previously analyzed SST patterns clarifies some important relationships with climate modes.

The breakdown into regional categories for the discussion with the emphasis on the modes of variability was sensible in the new framework. I was particularly gratified to see that the first panel of Figure 3 now makes the complicated patterns in the second panel of Figure 3 (formerly Figure 4 of the original manuscript, I think) much more comprehensible.

The tables, while complicated, convey important information and will be used to further understanding in all of the region sub-categories.

Response: We would like to again sincerely thank the reviewer for their very thoughtful and constructive reviews of our manuscript, which have led to substantial improvements in the structure, quality and readability of the paper.

My only minor comment is that it appears that the resolution of Figure 3 seems to be a bit grainy, higher resolution in the final version would be helpful.

Response: The Figure 3 resolution is consistent with the grid-scale of our analysis. We now provide Figure 3 in .pdf format with our revised manuscript.

I think the new version addresses all of the issues from the original review. The new organizing principle works well, and while the patterns and statistical relationships are still complex, the manuscript reads well.

Excellent job in improving the manuscript, and I find this acceptable for publication with the one minor comment on the figure in mind.

Response: Thank you.